# Practical Bayes-Optimal Membership Inference Attacks

**Marcus Lassila**[1]    **Johan Östman**[2]    **Khac-Hoang Ngo**[3]    **Alexandre Graell i Amat**[1]

[1]Chalmers University of Technology    [2]AI Sweden    [3]Linköping University

## Abstract

We develop practical and theoretically grounded membership inference attacks (MIAs) against both independent and identically distributed (i.i.d.) data and graph-structured data. Building on the Bayesian decision-theoretic framework of [1], we derive the Bayes-optimal membership inference rule for node-level MIAs against graph neural networks, addressing key open questions about optimal query strategies in the graph setting. We introduce BASE and G-BASE, tractable approximations of the Bayes-optimal membership inference. G-BASE achieves superior performance compared to previously proposed classifier-based node-level MIA attacks. BASE, which is also applicable to non-graph data, matches or exceeds the performance of prior state-of-the-art MIAs, such as LiRA and RMIA, at a significantly lower computational cost. Finally, we show that BASE and RMIA are equivalent under a specific hyperparameter setting, providing a principled, Bayes-optimal justification for the RMIA attack.

## 1 Introduction

Machine learning models are known to leak information about their training data [2–5], driving growing interest in understanding and quantifying such leakage. Due to the complexity of modern models, formally characterizing their information leakage remains a formidable challenge. As a result, privacy is often assessed empirically via so-called *privacy auditing*—designing attacks to identify data leakage and guide mitigation strategies.

Membership inference attacks (MIAs), where an adversary seeks to infer whether a specific data point was included in the training set of a model, represent the most fundamental privacy attack. Importantly, MIAs already pose a serious privacy threat: disclosing the mere presence of a data point in the training set may constitute a serious privacy violation and directly contravenes privacy regulations such as the GDPR. The European Data Protection Board explicitly cites MIAs in its guidance on when a machine learning model can be considered anonymous [6]. Moreover, MIAs can serve as a key component in data reconstruction attacks by filtering out unlikely candidates, thereby narrowing the search space [7], and can be used to establish lower bounds on the privacy guarantees of differentially-private algorithms [8].

State-of-the-art MIAs often rely on *shadow models*—auxiliary models trained under similar conditions as the target model—to empirically characterize behavioral differences between training and non-training data. Existing attacks fall into two main categories: classifier-based and statistic-based. Classifier-based attacks use features from shadow models, such as losses or logits, to train a binary classifier [2, 9, 10]. Statistic-based attacks instead compute statistical metrics using signals from both the target and shadow models [1, 11–13]. A Bayes-optimal strategy to membership inference was proposed in [1], but an exact computation is intractable, necessitating approximations. State-of-the-art methods such as LiRA [11] and RMIA [13], instead, are based on a hypothesis testing formula-

39th Conference on Neural Information Processing Systems (NeurIPS 2025).

tion [14]. While these approaches achieve strong empirical performance and outperform classifier-based attacks, their theoretical connection to the Bayes-optimal membership inference remains unclear.

Most existing work on MIAs assumes data points are independent and identically distributed (i.i.d.). Recently, however, MIAs have been studied in the context of graph-structured data, particularly against graph neural networks (GNNs). In this setting, message passing introduces structural dependencies—each node or edge can influence many others—making membership signals harder to isolate and challenging key assumptions underlying classical attacks. MIAs on graph-structured data can be grouped by the information they aim to recover: (i) node-level attacks attempt to infer whether a specific node was part of the training set [15–19]; (ii) edge-level attacks target the presence of specific edges [20–24]; and (iii) graph-level attacks aim to determine whether an entire graph instance was used in the

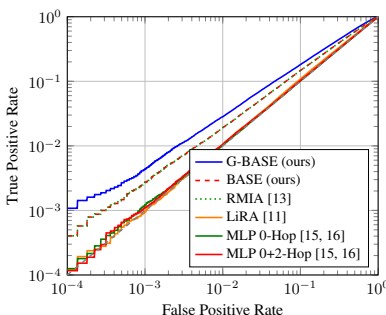

Figure 1: ROC curves of our attack and prior MIAs on the Flickr dataset, averaged over 10 GCN target models.

training [25–27]. At the node level, existing approaches are exclusively classifier-based. Given the success of statistic-based methods on the i.i.d. data, extending such strategies to graph data may lead to stronger attacks. However, this remains an open problem.

**Our contribution.** Motivated by the need for practical, theoretically grounded privacy auditing tools, we develop novel MIAs for both i.i.d. and graph data. Specifically, we design Bayes-optimal attacks building on the Bayesian decision-theoretic framework of [1], yielding attacks that are both effective and tractable. The proposed attacks tend to match or outperform the state-of-the-art, especially on larger datasets. Our key contributions include:

- We derive the Bayes-optimal decision rule for node-level membership inference on graph data, extending the results of Sablayrolles et al. [1] beyond the i.i.d. setting. This result formalizes how graph structure should be exploited in membership inference against GNNs.
- Guided by this Bayes-optimal rule, we propose G-BASE, a practical attack on GNNs that achieves state-of-the-art performance on graph data, significantly outperforming existing classifier-based methods.
- We propose BASE, a practical Bayes-optimal attack achieving state-of-the-art performance while being significantly more cost-efficient than prior methods such as RMIA [13] and LiRA [11].
- We reveal a close connection between BASE and RMIA: the two are equivalent (for a specific value of RMIA's threshold $\gamma$) up to a monotonically increasing transformation. However, BASE achieves the same performance with significantly lower computational cost, requiring much fewer model queries.

## 2 Related Work

**MIAs for i.i.d. data.** Relevant works include [2],[1], [14], [11] (LiRA), and [13] (RMIA). Shokri et al. [2] introduced classifier-based black-box MIAs: the adversary trains multiple shadow models on known datasets to mimic the target model, then trains a binary-classifier attack model on their outputs labeled by the ground truth training membership status. Sablayrolles et al. [1] proposed a Bayes-optimal framework in which membership inference amounts to computing the posterior probability that a data point is in the training set, given the trained model. Under mild assumptions, the optimal inference depends only on the loss, motivating black-box attacks where the attacker can observe only the model's output (e.g., the loss) on the target sample. Ye et al. [14] cast MIAs as a hypothesis testing game: a challenger samples a data point from either the training set or the rest of the data population (with equal probability), and an adversary, with access to the data population, must decide between two hypothesis—whether the model was trained on a dataset including or excluding the target point. State-of-the-art black-box attacks, such as LiRA [11] and RMIA [13], build on this formulation. LiRA performs a likelihood-ratio test between the two hypotheses, representing them via the distributions of losses (or logits) on the target data point. LiRA assumes the (transformed) logits exhibit Gaussian-like behavior, resulting in a parametric test. In RMIA, the test compares the case where the target data point is in the training set to the case where it is replaced by an auxiliary sample drawn randomly from the population. The test score is the tail probability of the resulting likelihood ratio.

**MIAs for graph data.** Existing node-level attacks [15–19] all adopt a classifier-based approach. These attacks involve three phases: training a shadow model, training a binary-classifier attack model, and performing membership inference. The attack model takes as input features the vector derived from the shadow model's outputs—either posterior probabilities [15–17, 19] or predicted labels [18]—for nodes that were included or not in the training set. Despite their promise, statistic-based methods have not yet been explored for node-level MIAs.

## 3 Preliminaries

**Graph notation.** For graph data, we consider a graph denoted by $\mathcal{G} = (\mathcal{V}, \mathcal{E}, \boldsymbol{X}, \boldsymbol{Y})$, where $\mathcal{V}$ is the set of $n$ nodes, $\mathcal{E}$ the set of edges, $\boldsymbol{X} \in \mathbb{R}^{n \times d}$ the node feature matrix, and $\boldsymbol{Y} \in \mathbb{R}^{n \times c}$ the one-hot encoded node-label matrix, such that each node $v \in \mathcal{V}$ has an associated feature-label pair $(\boldsymbol{x}_v, \boldsymbol{y}_v)$, which are assumed to be sensitive data. We denote by $\boldsymbol{A} \in \mathbb{R}^{n \times n}$ the adjacency matrix of graph $\mathcal{G}$, where $A_{uv} = 1$ if $(u, v) \in \mathcal{E}$ and 0 otherwise, and by $\mathcal{N}(v) = \{u : (u, v) \in \mathcal{E}, u \neq v\}$ the set of neighbors of node $v$. For convenience, in the analysis sections, we will refer to a graph $\mathcal{G}$ as $\mathcal{G} = (\boldsymbol{X}, \boldsymbol{Y}, \boldsymbol{A})$, where the sets of nodes and edges are implicitly defined by $\boldsymbol{X}$ and $\boldsymbol{A}$, respectively.

**Graph neural networks (GNNs).** GNNs [28–33], learn node embeddings that capture both structural and feature information. These embeddings are then used for various downstream tasks such as node classification. GNNs operate via message passing, where nodes aggregate information from their neighbors to update their embeddings. Formally, the embedding of node $v \in \mathcal{V}$ at level $\ell + 1$ of an $L$-layer GNN is computed as

$$\boldsymbol{h}_v^{(\ell+1)} = \text{UPDATE}\Big( \boldsymbol{h}_v^{(\ell)}, \text{AGGREGATE}\Big( \big\{ \boldsymbol{h}_u^{(\ell)}, u \in \mathcal{N}(v) \big\} \Big) \Big), \tag{1}$$

where both AGGREGATE, a permutation-invariant function, and UPDATE are differentiable and $\boldsymbol{h}_v^{(0)}$ is the input feature vector of node $v$. The computation of $\boldsymbol{h}_v^{(L)}$, the final embedding of node $v$, thus depends on its $L$-hop neighborhood, which we denote by $\mathcal{N}_L(v)$. For notational convenience, we denote the final embedding of node $v$ by $\boldsymbol{z}_v = \boldsymbol{h}_v^{(L)}$ and refer to it simply as the node embedding.

For node classification, a softmax operation is typically applied to the node embeddings to produce class probabilities,

$$P(\boldsymbol{y}_v | \boldsymbol{\theta}, \boldsymbol{X}, \boldsymbol{A}) \approx \text{softmax}(\boldsymbol{z}_v)_y \equiv f_{\boldsymbol{\theta}}(\boldsymbol{X}, \boldsymbol{A})_{vy} \tag{2}$$

where $f_{\boldsymbol{\theta}}(\boldsymbol{X}, \boldsymbol{A})_v$ denotes the softmax-normalized prediction vector for node $v$, computed by a GNN $f_{\boldsymbol{\theta}}$ with parameters $\boldsymbol{\theta}$, and $y$ denotes the index of the nonzero element of $\boldsymbol{y}_v$. The model is typically trained by minimizing the negative log-likelihood loss $\ell(f_{\boldsymbol{\theta}}(\boldsymbol{X}, \boldsymbol{A})_v, \boldsymbol{y}_v) = -\log P(\boldsymbol{y}_v | \boldsymbol{\theta}, \boldsymbol{X}, \boldsymbol{A})$ (i.e., the cross-entropy loss for node classification) over the labeled nodes, i.e., minimizing

$$\mathcal{L}(\boldsymbol{\theta}, \boldsymbol{X}, Y, \boldsymbol{A}) = \sum_{v \in \mathcal{V}} \ell(f_{\boldsymbol{\theta}}(\boldsymbol{X}, \boldsymbol{A})_v, \boldsymbol{y}_v) = -\sum_{v \in \mathcal{V}} \log P(\boldsymbol{y}_v | \boldsymbol{\theta}, \boldsymbol{X}, \boldsymbol{A}). \tag{3}$$

## 4 Bayes-Optimal Node-Level MIAs Against GNNs

We extend the Bayesian membership inference strategy in [1] to GNNs. We begin by framing the MIA as an indistinguishability game and then proceed to derive the Bayes-optimal decision rule for membership inference. Finally, we discuss tractable approximations and sampling strategies that yield powerful and practical MIAs.

### 4.1 Node Membership Inference Attack Game

Following [11, 14], we formulate the MIA as a game between a challenger and an adversary.

*Definition 1. (Membership inference game on graph data)*

*1. The challenger samples a subset of graph $\mathcal{G}$, $\mathcal{G}_{train} = (\mathcal{V}_{train}, \mathcal{E}_{train}) \subset \mathcal{G}$, where $\mathcal{V}_{train} \in \mathcal{V}$ and $\mathcal{E}_{train} = \{(u, v) : u, v \in \mathcal{V}_{train}, (u, v) \in \mathcal{E}\}$, and trains a GNN model $f_{\boldsymbol{\theta}}$ on $\mathcal{G}_{train}$. Let $m_v$ denote the membership status of node $v \in \mathcal{V}$,*

$$m_v = \begin{cases} 1, & v \in \mathcal{V}_{train} \\ 0, & v \notin \mathcal{V}_{train} \end{cases}. \tag{4}$$

2. *The challenger flips a fair coin to generate a bit b. If $b = 0$, it samples a target node $v$ from $\mathcal{G}\backslash\mathcal{G}_{train}$. Otherwise, it samples a target node $v$ from $\mathcal{G}_{train}$.*

3. *The challenger gives the adversary the full graph $\mathcal{G} = (\boldsymbol{X}, \boldsymbol{Y}, \boldsymbol{A})$, the target node $v$, and black-box access to the trained model $f_{\boldsymbol{\theta}}$.*

4. *The adversary may also have additional side information $\mathcal{H}$ (e.g., knowledge about the training algorithm or model architecture). Using this information, the adversary performs a MIA $\hat{m}_v \leftarrow MIA(f_{\boldsymbol{\theta}}, v, \mathcal{G}, \mathcal{H})$, where $\hat{m}_v$ is an estimate of the membership status of node $v$, $m_v$.*

5. *The attack is successful on a target node $v$ if $\hat{m}_v = m_v$.*

**Threat model.** We assume that the target graph $\mathcal{G}_{\text{train}} \subset \mathcal{G}$ is sampled from a larger, fixed graph $\mathcal{G}$ that is accessible to the adversary. This setup offers a flexible and practical way to model the data distribution. Notably, this corresponds to the *train on subgraph, test on full* setting introduced in [15]. When $\mathcal{G}$ is large relative to $\mathcal{G}_{\text{train}}$, this setup corresponds to the *data population pool* assumption widely adopted in membership inference [14, 13, 11]. While giving the adversary access to $\mathcal{G}$ simplifies the analysis of the Bayes-optimal attack, it also results in a stronger adversary. Nevertheless, this worst case assumption is suitable for privacy auditing, where conservative evaluations are preferred. We adopt a black-box setting, where the adversary has unlimited query access to the target model and can generate soft predictions for any valid input graph. Knowledge of the model's architecture, training algorithm, and objective function is encompassed in $\mathcal{H}$. In contrast, a white-box setting would grant the adversary direct access to model parameters or activations.

GNN training can be categorized into *transductive* and *inductive*. In the transductive setting, only part of the graph is labeled and used for loss computation. However, message-passing is performed over all nodes, producing an embedding for each node. The goal of this semi-supervised learning approach is typically to predict the labels of the unlabeled nodes. The inductive setting, on the other hand, corresponds to fully supervised learning. In this setting, message-passing is only performed between labeled nodes. The goal is then to generalize to unseen nodes. As noted in [16], the inductive setting is the most relevant, as it provides a clear distinction between member and non-member nodes. In contrast, the transductive setting—where the model is primarily used by the data holder to predict labels for unlabeled nodes and remains under their control—does not raise significant privacy concerns. We therefore focus our analysis and experiments on the inductive setting.

### 4.2 The Bayes-Optimal Decision Rule

Assume that the target node whose membership status we aim to infer is node $v \in \mathcal{I}$, and let $\mathcal{N}_L(v)$ be its $L$-hop neighborhood. Also, let $\mathcal{M} = \{m_u : u \in \mathcal{V}\}$ denote the membership indicator variables of all $n$ nodes in the graph $\mathcal{G}$, and $\tilde{\mathcal{M}} = \mathcal{M}\backslash\{m_v\}$ represent the membership statuses of all nodes except the target node $v$. Given the attack game and threat model defined in Section 4.1, a Bayesian adversary seeks to compute the posterior probability that the target node is in the training set, i.e., $P(m_v = 1|\boldsymbol{\theta}, \mathcal{G})$. The following theorem provides a closed-form expression for this posterior.

***Theorem 1.*** *Given a graph $\mathcal{G} = (\boldsymbol{X}, \boldsymbol{Y}, \boldsymbol{A})$ and an $L$-layer GNN model $\boldsymbol{\theta}$ trained on an induced subgraph of $\mathcal{G}$ to minimize the objective in* (3)*, the posterior probability $P(m_v = 1|\boldsymbol{\theta}, \mathcal{G})$ is given by*

$$P(m_v = 1|\boldsymbol{\theta}, \mathcal{G}) = \mathbb{E}_{\tilde{\mathcal{M}} \sim P(\tilde{\mathcal{M}}|\boldsymbol{\theta}, \mathcal{G})}\left[\sigma\left(-S_{\mathcal{L}}(f_{\boldsymbol{\theta}}, v, \tilde{\mathcal{M}}, \mathcal{G}) + \log\Lambda(\boldsymbol{\theta}, \tilde{\mathcal{M}}, \boldsymbol{X}, \boldsymbol{A})\right.\right.$$

$$\left.\left.-\log\int e^{-S_{\mathcal{L}}(f_{\boldsymbol{\phi}}, v, \tilde{\mathcal{M}}, \mathcal{G}) + \log\Lambda(\boldsymbol{\phi}, \tilde{\mathcal{M}}, \boldsymbol{X}, \boldsymbol{A})} p(\boldsymbol{\phi}|m_v = 0, \tilde{\mathcal{M}}, \mathcal{G})\mathrm{d}\boldsymbol{\phi} + \log\frac{\lambda}{1-\lambda}\right)\right], \quad (5)$$

*where*

$$S_{\mathcal{L}}(f_{\boldsymbol{\theta}}, v, \tilde{\mathcal{M}}, \mathcal{G}) = \ell(f_{\boldsymbol{\theta}}(\boldsymbol{X}, \boldsymbol{A}_{\mathcal{M}})_v, \boldsymbol{y}_v) + \Delta\mathcal{L}_{\mathcal{N}_L(v)}(f_{\boldsymbol{\theta}}), \quad (6)$$

*with*

$$\Delta\mathcal{L}_{\mathcal{N}_L(v)}(f_{\boldsymbol{\theta}}) = \sum_{u \in \mathcal{N}_L(v)} m_u(\ell(f_{\boldsymbol{\theta}}(\boldsymbol{X}, \boldsymbol{A}_{\mathcal{M}})_u, y_u) - \ell(f_{\boldsymbol{\theta}}(\boldsymbol{X}, \boldsymbol{A}_{\tilde{\mathcal{M}}})_u, y_u)), \quad (7)$$

*and*

$$\Lambda(\boldsymbol{\theta}, \tilde{\mathcal{M}}, \boldsymbol{X}, \boldsymbol{A}) = \frac{p(\boldsymbol{\theta}|m_v = 1, \tilde{\mathcal{M}}, \boldsymbol{X}, \boldsymbol{A})}{p(\boldsymbol{\theta}|m_v = 0, \tilde{\mathcal{M}}, \boldsymbol{X}, \boldsymbol{A})}, \quad (8)$$

*the latter representing the likelihood ratio prior to observing the labels. In* (5), $\lambda = P(m_v = 1)$ *denotes the prior probability that node $v$ is a member of the training set, before observing the model, and $\sigma$ is the sigmoid function. Also, the $n \times n$ matrices $\boldsymbol{A}_{\mathcal{M}}$ and $\boldsymbol{A}_{\tilde{\mathcal{M}}}$ in* (6) *and* (7) *are defined as*

$$(\boldsymbol{A}_{\mathcal{M}})_{uw} = \left\{ \begin{array}{ll} A_{uw}, & m_u = m_w = 1, \\ 0, & otherwise \end{array} \right. \quad and \quad (\boldsymbol{A}_{\tilde{\mathcal{M}}})_{uw} = \left\{ \begin{array}{ll} (\boldsymbol{A}_{\mathcal{M}})_{uw}, & u, w \neq v, \\ 0, & otherwise \end{array} \right.$$

*for $u, w \in \mathcal{V}$.*

*Proof.* See Appendix B.1. □

In (5), $S_{\mathcal{L}}(f_{\boldsymbol{\theta}}, v, \tilde{\mathcal{M}}, \mathcal{G})$ represents the loss-based signal for the target node $v$, while the first logarithmic term corresponds to the expected signal over the distribution of the models induced by the membership configuration $\tilde{\mathcal{M}}$, $p(\boldsymbol{\phi}|m_v = 0\tilde{\mathcal{M}}, \mathcal{G})$. The term $\Delta\mathcal{L}_{\mathcal{N}_L(v)}$ captures how the inclusion or exclusion of the target node influences the loss values of its neighbors (since an $L$-layer GNN aggregates information from nodes within the $L$-hop neighborhood, the optimal attack must consider this local graph structure around the target node, see Figure 2).

The presence of the term (7) in the graph data setting, in contrast to the i.i.d. case (see [1, Thm. 2] and Corollary 1 in Section 5.1), highlights a key distinction: dependencies between data points play a central role.

To make predictions, we set a decision threshold $\tau$ such that target node $v$ is inferred to be part of the training set if $P(m_v = 1|\boldsymbol{\theta}, \mathcal{G}) > \tau$. For auditing purposes, it is important to evaluate performance across the full range of false positive rates, particularly at low false positive rates, and we compute the receiver operating characteristic (ROC) curve by sweeping over $\tau \in [0, 1]$. Appendix C provides a detailed discussion on how an adversary can select the decision threshold $\tau$ by attacking simulated target models.

### 4.3 G-BASE: Practical Bayes-Optimal MIA against GNNs

The Bayes-optimal membership inference score function in Theorem 1 is computationally intractable and thus necessitates approximations. In this section, we introduce a practical and effective approximation to the Bayes-optimal decision rule, resulting in a powerful MIA.

First, the Bayesian membership inference involves computing a log-likelihood ratio of the membership indicator (see (12)). When attempting to approximate (5), we need to pick a prior (8) on the likelihood ratio, before observing the labels. A natural and simplifying approximation is to set $\Lambda(\boldsymbol{\theta}, \tilde{\mathcal{M}}, \boldsymbol{X}, \boldsymbol{A}) = 1$. Intuitively, distinguishing the distribution of models trained on node $v$ from the distribution of models not trained on $v$ is more difficult when the labels are unobserved, and $\log \Lambda(\boldsymbol{\theta}, \tilde{\mathcal{M}}, \boldsymbol{X}, \boldsymbol{A})$ small in comparison to the posterior likelihood ratio that we are estimating. Consequently, we would expect it to be small in comparison to the loss signal. The outcome is that the prior LLR term vanishes from (5).

The rest of the intractability stems from the two nested expectations in Equation (5): one over the membership statuses of all non-target samples, $\tilde{\mathcal{M}}$, and the other over the model distribution, $p(\boldsymbol{\phi}|m_v = 0, \tilde{\mathcal{M}}, \mathcal{G})$. The expectation over $\tilde{\mathcal{M}}$ requires sampling from the conditional distribution $P(\tilde{\mathcal{M}}|\boldsymbol{\theta}, \mathcal{G})$, which is infeasible (it is essentially the joint membership inference problem over all non-target nodes). To address this, we propose three approximate sampling strategies to generate samples from $P(\tilde{\mathcal{M}}|\boldsymbol{\theta}, \mathcal{G})$: i) *model-independent sampling*, which assumes independence from the target model, i.e., $P(\tilde{\mathcal{M}}|\boldsymbol{\theta}, \mathcal{G}) \approx P(\tilde{\mathcal{M}}|\mathcal{G})$; ii) 0-*hop MIA sampling*, which leverages a MIA attack for the i.i.d. setting to obtain per-node membership probabilities; and iii) *Markov chain Monte Carlo sampling*, which accounts explicitly for the dependence on the target model $\boldsymbol{\theta}$. These strategies are discussed in greater detail in Appendix B.2. Given any of these sampling strategies, we generate $M$ samples $\tilde{\mathcal{M}}_1, \dots, \tilde{\mathcal{M}}_M$ and approximate the outer expectation $\mathbb{E}_{\tilde{\mathcal{M}} \sim P(\tilde{\mathcal{M}}|\boldsymbol{\theta}, \mathcal{G})}[\cdot]$ in (5) with the sample average.

The remaining expectation over the model distribution $p(\boldsymbol{\phi}|m_v = 0, \tilde{\mathcal{M}}, \mathcal{G})$ (the integral term in (5)) is also intractable. We approximate it via Monte Carlo sampling using a set of shadow models trained on subgraphs of $\mathcal{G}$, interpreting $p(\boldsymbol{\phi}|m_v = 0, \tilde{\mathcal{M}}, \mathcal{G})$ as the distribution of models trained on the subgraph defined by $\tilde{\mathcal{M}}$ with the target node excluded. However, directly sampling from this

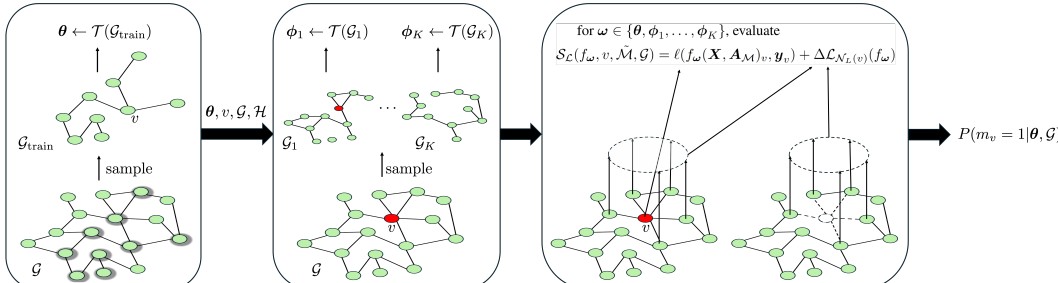

Figure 2: Visualization of the Bayes-optimal attack. From left to right: the challenger trains the target model $\boldsymbol{\theta}$ on $\mathcal{G}_{\text{train}}$, and provides the trained model, a target node $v$, the underlying graph $\mathcal{G}$, and (optionally) auxiliary information $\mathcal{H}$ detailing the training procedure. The adversary then samples $K$ graphs and trains a corresponding set of shadow models $\{\phi_i\}_{i=1}^K$. These sampled graphs may or may not contain the target node $v$. Finally, the adversary estimates the membership of $v$ using the Bayes-optimal decision rule in (5), approximated via the Monte Carlo method in (9).

distribution would require training a separate set of shadow models for each pair $(v, \tilde{\mathcal{M}})$, which is computationally infeasible. For $N$ target nodes, $M$ samples of $\tilde{\mathcal{M}}$, and $K$ shadow models per configuration, this would amount to training $N \times M \times K$ models. To reduce the number of shadow models, we approximate $p(\phi | m_v = 0, \tilde{\mathcal{M}}, \mathcal{G})$ by a distribution $p(\phi | \mathcal{G})$ only dependent on the data population. This simplification allows us to reuse the same set of shadow models across all targets and sampled membership configurations $\tilde{\mathcal{M}}$. More precisely, the adversary samples subgraphs $\mathcal{G}_1, \dots, \mathcal{G}_K$ from $\mathcal{G}$ and use them to train shadow models. Ideally, the adversary should sample subgraphs and train shadow models using a similar procedure as the challenger. However, this distribution is constrained by the adversary's side information.

The integral in (5) can then be approximated as

$$
\log \int e^{-S_\mathcal{L}(f_\phi, v, \tilde{\mathcal{M}}, \mathcal{G})} p(\phi | m_v = 0, \tilde{\mathcal{M}}, \mathcal{G}) \mathrm{d}\phi \approx \log\left( \frac{1}{K} \sum_{k=1}^{K} e^{-S_\mathcal{L}(f_{\phi_k}, v, \tilde{\mathcal{M}}, \mathcal{G})} \right), \quad \phi_k \leftarrow \mathcal{T}(\mathcal{G}_k),
$$
(9)

where $\mathcal{T}$ is the training algorithm.

Incorporating both approximations into (5), we arrive at the following attack:

**Definition 2. (G-BASE attack)** *For a given threshold $\tau$, and a set of shadow models $\{\phi_k\}_{k=1}^K$, target sample $v$ is inferred to be part of the training set if $P(m_v = 1 | \boldsymbol{\theta}, \mathcal{G}) > \tau$, where*

$$
P(m_v = 1 | \boldsymbol{\theta}, \mathcal{G}) \approx \frac{1}{M} \sum_{i=1}^{M} \sigma\left( -S_\mathcal{L}(f_{\boldsymbol{\theta}}, v, \tilde{\mathcal{M}}_i, \mathcal{G}) - \log\left( \frac{1}{K} \sum_{k=1}^{K} e^{-S_\mathcal{L}(f_{\phi_k}, v, \tilde{\mathcal{M}}_i, \mathcal{G})} \right) + \log \frac{\lambda}{1 - \lambda} \right)
$$

We refer to this attack as the graph Bayes-approximate membership status estimation (G-BASE) attack. An illustration of the G-BASE attack is shown in Figure 2. The distinction between online and offline variants is discussed in Appendix F.

## 5 Bayes-Optimal MIAs for i.i.d. Data

In this section, we consider MIAs in the i.i.d. setting. We first show that the formulation in (5)–(8) recovers the result for the i.i.d. case presented in [1] (Theorem 2) as a special case. We then introduce a practical approximation of the Bayes-optimal decision rule that yields a powerful attack. The formal definition of the membership inference game in the i.i.d. setting is provided in Appendix D.

### 5.1 The Bayes-Optimal Decision Rule

The result below follows as a corollary of Theorem 1.

**Corollary 1.** *Let* $\mathcal{D} = \{(\boldsymbol{x}_v, \boldsymbol{y}_v)\}_{v=1}^n$ *be a set of $n$ i.i.d. data samples and denote* $t_\lambda = \log \frac{\lambda}{1-\lambda}$. *For i.i.d. data, assuming the prior* $p(\boldsymbol{\theta}|m_v, \tilde{\mathcal{M}}, \{\boldsymbol{x}_v\}_{v=1}^n)$ *is independent of $m_v$, (5)–(8) reduces to*

$$P(m_v = 1|\boldsymbol{\theta}, \mathcal{D})$$
$$= \mathbb{E}_{\tilde{\mathcal{M}} \sim P(\tilde{\mathcal{M}}|\boldsymbol{\theta}, \mathcal{D})} \left[ \sigma \left( -\ell(f_{\boldsymbol{\theta}}(\boldsymbol{x}_v), \boldsymbol{y}_v) - \log \int e^{-\ell(f_{\boldsymbol{\theta}}(\boldsymbol{x}_v), \boldsymbol{y}_v)} p(\boldsymbol{\phi}|m_v = 0, \tilde{\mathcal{M}}, \mathcal{D}) \mathrm{d}\boldsymbol{\phi} + t_\lambda \right) \right],$$

*i.e., the result in [1, Thm. 2] with the temperature parameter absorbed into the loss function.*

*Proof.* See Appendix E.1. □

## 5.2 BASE: Practical Bayes-Optimal MIA for i.i.d. Data

As for the graph case, the Bayes-optimal decision rule stated in Theorem 1 and in [1, Thm. 2] (for i.i.d. data) is computationally intractable. For the i.i.d. case, [1] proposed several approximations; however, the resulting attacks underperform compared to those in [11, 13]. In this section, we introduce an alternative approximation to the Bayes-optimal decision rule for the i.i.d. setting that yields a practical and powerful attack.

Due to space constraints, we present the formal definition of the proposed attack below and refer the reader to Appendix E.2 for further details.

**Definition 3.** *(BASE attack) For a given threshold $\tau$, and a set of shadow models $\{\boldsymbol{\phi}_k\}_{k=1}^K$, target sample $v$ is inferred to be part of the training set if $P(m_v = 1|\boldsymbol{\theta}, \mathcal{D}) > \tau$, where*

$$P(m_v = 1|\boldsymbol{\theta}, \mathcal{D}) = \sigma \left( -\ell(f_{\boldsymbol{\theta}}(\boldsymbol{x}_v), \boldsymbol{y}_v) - \log \left( \frac{1}{K} \sum_{k=1}^K e^{-\ell(f_{\boldsymbol{\phi}_k}(\boldsymbol{x}_v), \boldsymbol{y}_v)} \right) + \log \frac{\lambda}{1-\lambda} \right). \quad (10)$$

We refer to this attack as the Bayes-approximate membership status estimation (BASE) attack.

As shown in Section 6 and Appendix H, despite relying on a coarse approximation of the Bayes-optimal decision rule in (5), the BASE attack matches or exceeds the performance of state-of-the-art attacks [11, 13].

**Connection to RMIA**. We formally establish a connection between the BASE attack and the RMIA attack proposed in [13]. To this end, we introduce a notion of equivalence for score-based MIAs (i.e. any MIA that produce soft membership scores that are thresholded into hard membership predictions).

**Definition 4.** *(MIA equivalence) Two score-based MIAs are said to be equivalent if, for any decision threshold for one attack, there exists a decision threshold for the other attack that yields identical hard predictions.*

In particular, equivalent attacks produce identical ROC curves.

The following theorem shows that the BASE attack is closely related to the RMIA attack [13].

**Theorem 2.** *The BASE attack is equivalent (in the sense of Definition 4) to RMIA when $\gamma = 1$. More precisely, the membership prediction scores produced by RMIA with $\gamma = 1$ is related to those of BASE via a monotone increasing function.*

*Proof.* See Appendix E.3. □

This result provides a theoretical justification for viewing RMIA as a Bayes-optimal attack, up to the approximation on the shadow model distribution.

Despite the equivalence between BASE and RMIA (in the sense of Definition 4), BASE offers significantly improved efficiency. In RMIA, the target model and shadow models need to be queried for each target sample and a sufficiently large number of population samples. In a practical implementation, a sampled subset $Z \sim \mathcal{D}$ is typically used in place of the full population $\mathcal{D}$ for efficiency, which can degrade performance [13]. BASE, on the other hand, only needs to query the models using the target samples to match the performance of RMIA with $Z = \mathcal{D}$, which is also the best performing configuration of RMIA, see [13, Table 4].

# 6 Experiments

We evaluate the attack performance of BASE and G-BASE across a range of datasets and model architectures. This section focuses on attacks against GNNs; results on i.i.d. data (CIFAR-10 and CIFAR-100) are presented in Appendix H.4. Open source code is available to reproduce our results.[1]

**Setup.** Experiments are conducted on 6 graph datasets: Cora, Citeseer, Pubmed, Flickr, Amazon-Photo, and Github. Cora, Citeseer and Pubmed are citation networks previously used in node-level MIA work [15–17]. For both the target and shadow models, we consider 2-layer GNN architectures: GCN [30], GRAPHSAGE with max aggregation [31], and GAT [32]. The models are trained inductively on randomly-induced subgraphs containing 50% of the nodes, and 50% of the dataset is used as target samples, evenly split between members and non-members.[2] Attack performance is measured in terms of the area under the receiver operating characteristic curve (AUC), and true positive rate (TPR) at 1% and 0.1% false positive rate (FPR). The AUC captures the average attack performance across all FPRs, while the attack performance at low FPR is most relevant in practice, as high TPR at a low FPR is essential for confident and reliable membership inference [11]. Additional details of the training procedure and hyperparameter selection are provided in Appendix A.

**Baseline attacks.** We compare the performance of BASE and G-BASE against LiRA [11] and RMIA [13], the current state-of-the-art MIAs for i.i.d. data. We also compare against node-level MLP-classifier attacks against GNNs from prior work [15–17]. To strengthen these baselines, we enhance the MLP-classifier attacks by using multiple shadow models to generate attack features, following the shadow training framework of [2]. The attacks are implemented based on the descriptions in their respective papers, incorporating information from the authors' code when available.

**Online and offline setting.** We evaluate both online and offline attacks, as defined in Appendix F, and compare BASE and G-BASE with LiRA and RMIA in both settings. Since MLP-classifier attacks use shadow models solely to generate training data for the MLP attack model, rather than to estimate reference model distributions, they are insensitive to whether the target nodes are included in any of the shadow models. In our setup, the shadow models are trained partially on target samples; therefore, the MLP-classifier attacks are formally considered online attacks.

**Attacks against GNNs.** Figure 1 shows ROC curves for G-BASE (using model-independent sampling) and several baselines on Flickr, using a 2-layer GCN architecture and 8 shadow models in online mode. G-BASE outperforms other baselines across the full range of FPRs on a majority of the benchmarks, especially on the larger datasets. Table 1 shows the performance of BASE and G-BASE and the baseline MIAs across multiple datasets and GNN target models. All attacks use the same set of $K$ shadow models for a fair comparison. The number of sampled graphs $M$ in G-BASE is set to 8. For $K = 8$ (online) and $K = 4$ (offline), we evaluate G-BASE for model-independent sampling (MI), 0-hop MIA sampling (MIA), and Gibbs sampling (see Section 4.3), using BASE as the 0-hop MIA. In the large $K$ experiments, we use 0-hop MIA sampling in all cases except for Flicker where we use model-independent sampling. Further results on the sampling strategy are presented and analyzed in Appendix H.2. The MLP-classifier attack is evaluated in two variants: using 0-hop query outputs as attack features, and using concatenated 0-hop and 2-hop query outputs to form attack features (0+2-hop). The hyperparameters for offline BASE and offline RMIA are selected using Bayesian optimization [34], with cross-validation over shadow models acting as simulated target models.

We make the following key observations: The MLP-classifier attacks are not competitive with our attacks BASE and G-BASE, or LiRA and RMIA across all datasets and target models and perform as random guessing on the larger, more challenging datasets. For $K = 4$ and 8 shadow models, BASE and G-BASE consistently outperform all baselines on the larger and more challenging datasets Pubmed, Flickr, Amazon-Photo, and Github. On these datasets, G-BASE effectively leverages local neighborhood information of the target node to enhance the attack performance. On the smaller datasets Cora and Citeseer ($\approx$ 3000 nodes out of which half is used to train the models), LiRA achieves comparatively good performance in terms of AUC. However, BASE, and G-BASE achieve superior performance in the more relevant low-FPR regime. Similarly, on Amazon-Photo ($\approx$ 7500 nodes), which is still smaller than the remaining datasets whose sizes range from approximately 20K to 90K nodes, the performance of LiRA is comparable to that of BASE, G-BASE, and RMIA. As the dataset size increases, however, LiRA's performance is overtaken by the other methods, indicating

---

[1]`https://github.com/MarcusLassila/MIA-audit-GNN`
[2]On Flickr and Github, only 20% are used as target samples when using 128 or 64 shadow models.

Table 1: Comparison of different attacks across datasets and model architectures. Performance is measured in terms of AUC and TPR at 1% and 0.1% FPR. The result is reported as the sample mean $\pm$ standard deviation over 10 random target models and samples of target nodes. The parameter $K$ denotes the number of shadow models. The attacks are evaluated in online mode for $K = 8$ and $K = 128$ and otherwise offline mode. Our attacks BASE and G-BASE achieves top performance in the majority of cases, sometimes with a significant margin.

| K | Attack | Cora (GCN) AUC (%) | TPR@FPR (%) 1% | 0.1% | Citeseer (GAT) AUC (%) | TPR@FPR (%) 1% | 0.1% | Pubmed (GraphSAGE) AUC (%) | TPR@FPR (%) 1% | 0.1% |
|---|---|---|---|---|---|---|---|---|---|---|
| 8 | MLP (0-hop) | 69.84 ± 1.74 | 3.83 ± 1.32 | 0.56 ± 0.51 | 73.15 ± 2.11 | 5.50 ± 1.34 | 0.97 ± 0.82 | 50.29 ± 0.58 | 1.10 ± 0.22 | 0.13 ± 0.07 |
|  | MLP (0+2-hop) | 70.14 ± 2.74 | 3.96 ± 1.65 | 0.93 ± 0.82 | 75.39 ± 2.44 | 7.71 ± 1.50 | 2.12 ± 1.40 | 51.20 ± 0.56 | 1.16 ± 0.25 | 0.12 ± 0.05 |
|  | LiRA | 82.35 ± 1.30 | 8.01 ± 4.42 | 1.02 ± 1.55 | **85.89 ± 0.63** | 13.38 ± 3.59 | 1.66 ± 1.73 | 52.77 ± 0.50 | 1.17 ± 0.14 | 0.12 ± 0.05 |
|  | RMIA | **82.45 ± 1.45** | 17.44 ± 1.72 | 4.64 ± 2.25 | 84.84 ± 1.13 | 22.23 ± 3.62 | 5.50 ± 4.45 | 57.44 ± 0.56 | 3.07 ± 0.51 | 0.55 ± 0.10 |
|  | BASE | **82.45 ± 1.45** | **17.47 ± 1.73** | 4.64 ± 2.25 | 84.84 ± 1.13 | **22.29 ± 3.64** | 5.51 ± 4.43 | 57.44 ± 0.56 | 3.07 ± 0.51 | 0.55 ± 0.10 |
|  | G-BASE (MI) | 77.15 ± 1.51 | 7.19 ± 3.08 | 0.68 ± 0.92 | 81.78 ± 0.67 | 13.62 ± 2.52 | 1.44 ± 0.89 | 62.92 ± 0.66 | 5.38 ± 0.49 | 1.20 ± 0.35 |
|  | G-BASE (MIA) | 77.42 ± 1.40 | 15.36 ± 3.04 | **5.05 ± 4.60** | 83.31 ± 0.78 | 21.19 ± 4.27 | **6.92 ± 4.61** | 62.96 ± 0.74 | 5.58 ± 1.03 | 1.16 ± 0.49 |
|  | G-BASE (Gibbs) | 76.86 ± 1.61 | 14.02 ± 2.51 | 4.40 ± 2.84 | 84.39 ± 0.67 | 21.38 ± 4.16 | 5.26 ± 3.80 | **64.10 ± 0.69** | **6.28 ± 0.56** | **1.85 ± 0.59** |
| 4 | LiRA | **82.37 ± 1.71** | 12.38 ± 6.09 | 2.10 ± 1.84 | 84.68 ± 1.05 | 18.39 ± 2.62 | 5.01 ± 2.95 | 55.40 ± 0.66 | 1.57 ± 0.25 | 0.18 ± 0.14 |
|  | RMIA | 80.57 ± 1.52 | 17.67 ± 1.69 | **7.87 ± 3.66** | 83.53 ± 1.29 | **26.70 ± 3.39** | **12.26 ± 5.19** | 56.73 ± 0.46 | 2.95 ± 0.41 | 0.57 ± 0.23 |
|  | BASE | 81.47 ± 1.14 | **17.87 ± 2.43** | 7.12 ± 3.99 | **84.83 ± 1.31** | 25.28 ± 3.22 | 10.37 ± 3.93 | 56.82 ± 0.60 | 3.02 ± 0.44 | 0.62 ± 0.24 |
|  | G-BASE (MI) | 74.25 ± 1.79 | 9.41 ± 2.33 | 3.03 ± 1.67 | 76.26 ± 1.24 | 13.92 ± 2.27 | 4.80 ± 3.24 | 61.97 ± 0.66 | 5.22 ± 0.38 | 1.35 ± 0.30 |
|  | G-BASE (MIA) | 73.60 ± 1.38 | 10.38 ± 2.97 | 3.53 ± 2.06 | 76.65 ± 1.33 | 15.22 ± 2.34 | 4.02 ± 2.36 | 62.06 ± 0.64 | 5.33 ± 0.47 | 1.27 ± 0.26 |
|  | G-BASE (Gibbs) | 73.83 ± 1.74 | 10.21 ± 2.68 | 3.10 ± 1.49 | 77.08 ± 1.19 | 14.69 ± 1.57 | 4.92 ± 3.46 | **62.58 ± 0.64** | **5.59 ± 0.42** | **1.58 ± 0.31** |
| 128 | LiRA | **89.49 ± 1.07** | **34.62 ± 5.63** | **19.51 ± 4.99** | **91.88 ± 0.74** | **47.14 ± 3.17** | **29.77 ± 6.14** | 56.40 ± 0.44 | 3.07 ± 0.40 | 0.73 ± 0.21 |
|  | RMIA | 83.28 ± 1.22 | 20.10 ± 2.31 | 7.86 ± 4.36 | 85.59 ± 1.25 | 25.27 ± 3.32 | 7.23 ± 4.36 | 57.58 ± 0.59 | 3.23 ± 0.45 | 0.58 ± 0.14 |
|  | BASE | 83.28 ± 1.22 | 20.10 ± 2.31 | 7.86 ± 4.36 | 85.59 ± 1.25 | 25.27 ± 3.32 | 7.23 ± 4.36 | 57.58 ± 0.59 | 3.23 ± 0.45 | 0.58 ± 0.14 |
|  | G-BASE | 77.21 ± 1.34 | 17.40 ± 5.39 | 7.33 ± 4.48 | 84.41 ± 0.85 | 25.51 ± 3.65 | 7.81 ± 3.19 | **63.09 ± 0.45** | **5.44 ± 0.69** | **1.18 ± 0.35** |
| 64 | LiRA | **86.39 ± 1.17** | **33.10 ± 5.45** | **19.35 ± 5.42** | **88.11 ± 0.92** | **45.15 ± 2.95** | **26.70 ± 5.64** | 56.34 ± 0.77 | 2.12 ± 0.41 | 0.42 ± 0.24 |
|  | RMIA | 81.38 ± 1.18 | 20.69 ± 2.98 | 11.60 ± 4.02 | 84.54 ± 1.30 | 31.22 ± 2.97 | 16.90 ± 3.18 | 57.44 ± 0.52 | 3.46 ± 0.54 | 0.86 ± 0.29 |
|  | BASE | 82.49 ± 1.18 | 21.85 ± 3.66 | 11.11 ± 4.06 | 85.98 ± 1.15 | 32.35 ± 3.10 | 15.82 ± 4.93 | 57.58 ± 0.62 | 3.55 ± 0.49 | 0.91 ± 0.37 |
|  | G-BASE | 74.88 ± 1.13 | 13.62 ± 2.58 | 4.34 ± 2.63 | 79.23 ± 1.01 | 21.12 ± 1.67 | 8.63 ± 4.41 | **63.62 ± 0.55** | **6.49 ± 0.36** | **2.09 ± 0.45** |

| K | Attack | Flickr (GCN) AUC (%) | TPR@FPR (%) 1% | 0.1% | Amazon-Photo (GAT) AUC (%) | TPR@FPR (%) 1% | 0.1% | Github (GraphSAGE) AUC (%) | TPR@FPR (%) 1% | 0.1% |
|---|---|---|---|---|---|---|---|---|---|---|
| 8 | MLP (0-hop) | 50.20 ± 0.24 | 1.02 ± 0.11 | 0.12 ± 0.03 | 52.11 ± 1.08 | 1.56 ± 0.49 | 0.27 ± 0.19 | 49.99 ± 0.38 | 0.91 ± 0.35 | 0.11 ± 0.07 |
|  | MLP (0+2-hop) | 50.35 ± 0.31 | 1.06 ± 0.08 | 0.11 ± 0.03 | 52.20 ± 1.28 | 1.62 ± 0.29 | 0.22 ± 0.16 | 50.01 ± 0.44 | 0.94 ± 0.07 | 0.09 ± 0.04 |
|  | LiRA | 51.63 ± 0.26 | 1.04 ± 0.12 | 0.10 ± 0.03 | 52.33 ± 0.86 | 2.23 ± 0.21 | 0.30 ± 0.21 | 51.27 ± 0.28 | 1.05 ± 0.08 | 0.11 ± 0.04 |
|  | RMIA | 56.23 ± 0.57 | 1.92 ± 0.21 | 0.27 ± 0.05 | 56.35 ± 0.93 | 2.19 ± 0.66 | 0.36 ± 0.22 | 54.13 ± 0.59 | 2.05 ± 0.31 | 0.34 ± 0.09 |
|  | BASE | 56.23 ± 0.57 | 1.92 ± 0.21 | 0.27 ± 0.05 | 56.35 ± 0.93 | 2.20 ± 0.67 | 0.36 ± 0.23 | 54.13 ± 0.59 | 2.05 ± 0.31 | 0.34 ± 0.09 |
|  | G-BASE (MI) | **60.04 ± 1.01** | **2.82 ± 0.30** | **0.44 ± 0.06** | **56.89 ± 0.45** | 3.70 ± 0.83 | 0.79 ± 0.38 | 57.40 ± 1.09 | 2.93 ± 0.44 | 0.49 ± 0.15 |
|  | G-BASE (MIA) | 57.48 ± 0.55 | 2.51 ± 0.19 | 0.39 ± 0.09 | 56.77 ± 0.85 | 3.99 ± 0.62 | **0.82 ± 0.47** | 57.38 ± 1.40 | 3.00 ± 0.52 | **0.55 ± 0.18** |
|  | G-BASE (Gibbs) | 57.88 ± 0.91 | 2.33 ± 0.27 | 0.37 ± 0.15 | 56.53 ± 0.64 | **4.44 ± 0.60** | 0.76 ± 0.49 | **57.47 ± 1.43** | **3.18 ± 0.54** | **0.55 ± 0.17** |
| 4 | LiRA | 55.71 ± 0.61 | 1.58 ± 0.12 | 0.17 ± 0.05 | 56.71 ± 1.24 | 3.17 ± 0.51 | 0.47 ± 0.27 | 53.16 ± 0.58 | 1.35 ± 0.18 | 0.16 ± 0.07 |
|  | RMIA | 56.15 ± 0.49 | 2.03 ± 0.20 | 0.32 ± 0.08 | 56.21 ± 1.07 | 3.55 ± 0.81 | 1.06 ± 0.52 | 53.98 ± 0.60 | 2.05 ± 0.22 | 0.36 ± 0.07 |
|  | BASE | 56.18 ± 0.50 | 2.05 ± 0.22 | 0.29 ± 0.04 | 56.61 ± 1.06 | 3.58 ± 0.85 | 0.88 ± 0.68 | 53.97 ± 0.61 | 2.07 ± 0.22 | 0.36 ± 0.10 |
|  | G-BASE (MI) | **60.04 ± 1.01** | **2.88 ± 0.26** | **0.50 ± 0.06** | 57.81 ± 0.71 | **4.94 ± 0.78** | 1.63 ± 0.44 | 57.78 ± 1.22 | 3.10 ± 0.56 | 0.58 ± 0.15 |
|  | G-BASE (MIA) | 57.71 ± 0.66 | 2.48 ± 0.21 | 0.40 ± 0.07 | 57.74 ± 0.85 | 4.82 ± 0.88 | 1.38 ± 0.37 | 57.72 ± 1.06 | 3.12 ± 0.47 | 0.60 ± 0.19 |
|  | G-BASE (Gibbs) | 58.63 ± 0.95 | 2.48 ± 0.32 | 0.37 ± 0.07 | **57.94 ± 0.54** | 4.61 ± 1.00 | **1.69 ± 0.82** | **58.16 ± 1.48** | **3.44 ± 0.64** | **0.73 ± 0.26** |
| 128 | LiRA | 54.87 ± 0.35 | 1.95 ± 0.20 | 0.21 ± 0.09 | **58.73 ± 0.91** | **6.43 ± 0.77** | **3.52 ± 0.47** | 53.68 ± 1.01 | 2.01 ± 0.43 | 0.28 ± 0.16 |
|  | RMIA | 56.00 ± 0.35 | 2.25 ± 0.38 | 0.36 ± 0.17 | 56.52 ± 0.76 | 2.67 ± 0.83 | 0.58 ± 0.33 | 54.66 ± 0.89 | 2.18 ± 0.41 | 0.38 ± 0.17 |
|  | BASE | 56.00 ± 0.35 | 2.25 ± 0.38 | 0.36 ± 0.17 | 56.52 ± 0.76 | 2.67 ± 0.83 | 0.58 ± 0.33 | 54.66 ± 0.89 | 2.18 ± 0.41 | 0.38 ± 0.17 |
|  | G-BASE | **58.94 ± 0.81** | **2.87 ± 0.38** | **0.44 ± 0.13** | 54.88 ± 0.98 | 3.82 ± 0.74 | 0.93 ± 0.51 | **57.02 ± 1.58** | **2.96 ± 0.56** | **0.51 ± 0.15** |
| 64 | LiRA | 56.13 ± 0.36 | 1.98 ± 0.27 | 0.23 ± 0.10 | **57.82 ± 1.12** | **5.64 ± 0.92** | **2.65 ± 0.53** | 54.07 ± 0.78 | 1.39 ± 0.31 | 0.17 ± 0.11 |
|  | RMIA | 56.12 ± 0.32 | 2.28 ± 0.23 | 0.41 ± 0.14 | 56.46 ± 1.03 | 3.95 ± 0.77 | 1.58 ± 0.43 | 54.76 ± 0.85 | 2.47 ± 0.43 | 0.49 ± 0.21 |
|  | BASE | 56.12 ± 0.33 | 2.28 ± 0.24 | 0.41 ± 0.14 | 56.95 ± 1.07 | 4.13 ± 0.78 | 1.46 ± 0.68 | 54.80 ± 0.83 | 2.50 ± 0.41 | 0.55 ± 0.23 |
|  | G-BASE | **59.66 ± 0.71** | **3.14 ± 0.17** | **0.52 ± 0.09** | 56.74 ± 0.72 | 4.92 ± 0.67 | 1.71 ± 0.58 | **57.99 ± 1.67** | **3.16 ± 0.62** | **0.77 ± 0.24** |

that the proposed approaches scale more effectively to larger graphs. As predicted by Theorem 2, in the online setting, BASE and RMIA yield nearly identical results. Minor differences stem from RMIA using only half of the nodes in the $Z$ set; including all nodes in the $Z$ set, makes the attacks equivalent, as established in Theorem 2. However, as noted in 5.2, BASE requires significantly less computation in terms of model queries, compared to RMIA. In fact, BASE requires only a single query of the target and shadow models, whereas RMIA also need to query the models over the $Z$ set. In the case of a large number of shadow models, the performance of LiRA increases significantly. On some datasets, most notably Cora and Citeseer, LiRA performs very well. However, on larger datasets, e.g. Flickr and Pubmed, the performance gain of LiRA is more modest. Our attacks BASE and G-BASE are more principled and robust against datasets and models, and achieve top performance in many settings and also for a large number of shadow models $K$. However, we consider attacks that rely on a large number of shadow model to be of less practical relevance, since training a large number of shadow models is often infeasible for large models and large real-world datasets.

**Comparing online and offline attacks.** Since the same target models and target nodes are used to evaluate all attacks for a given dataset and GNN architecture in Table 1, the online and offline attacks are directly comparable. Despite using only half the number of shadow models, there are several cases where offline BASE and G-BASE is outperforming its online counterpart. This trend is most clear when comparing the large $K$ attacks, presumably because of the diminishing returns of increasing the number of shadow models from 64 to 128. In light of Theorem 1, this is indeed what we would expect since the target node is explicitly absent from the distribution of reference models. Furthermore, the added hyperparameter in the offline versions of RMIA and BASE is likely another contributing factor to this phenomenon.

**Robustness to mismatched adversary assumptions.** We evaluate and compare the attacks in a more challenging setting where the adversary lacks precise knowledge of the challenger's training procedure and model architecture. Specifically, the challenger trains a 2-layer GCN on 35% of Cora, while the adversary trains 8 2-layer GAT shadow models on 50% of the nodes, following the shadow model training procedure outlined in Appendix A. Furthermore, the challenger trains the shadow model using a SGD optimizer with momentum, while the adversary uses an Adam optimizer.

Figure 3 presents the results of our attacks and baseline attacks (see Appendix H.1 for more results). In this setting, all attacks show degraded performance, with AUC scores dropping by around 10 percentage points or more across all attacks, compared to the performance against a GCN target model on Cora (see Table 1 and Table 8). LiRA suffers the most, losing around 15 AUC percentage points and performing worse than MLP-classifier attacks in the low FPR regime. In contrast, our attacks and RMIA are more robust to mismatches in model and training procedures introduced by the adversary, achieving the highest TPR at low FPRs.

**Computational aspects.** An advantage of BASE over prior state-of-the-art attacks LiRA and RMIA is its improved efficiency. In Section 5.2, in light of the equivalence between BASE and RMIA, we discussed how BASE requires much fewer model queries compared to RMIA. In addition to advantages such as being more stealthy and robust to imposed limits on allowed queries, it also improves the computational efficiency of the inference phase of the attack.

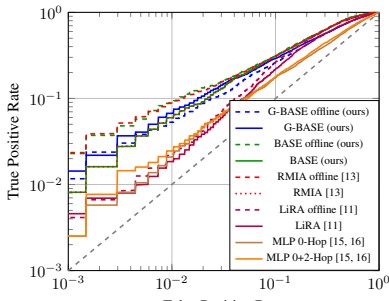

Figure 3: ROC of a mismatched attack averaged over 10 independent target models. Shadow models utilize a GAT architecture and different training procedure. 8 shadow models for online; 4 for offline.

Compared to LiRA, there is no advantage in terms of model queries since LiRA also only requires querying the models with the target samples. However, LiRA typically requires a larger number of shadow models to reach its full potential as our results indicate (see Table 1 and Table 12), and which is also found in [13]. Training the shadow models is the most costly part, especially for larger models and datasets. By achieving good performance at a lower number of shadow models, BASE has a computational performance advantage over LiRA.

As for G-BASE, although it is a tractable approximation of the Bayes-optimal membership inference on graph data, it is less efficient than the 0-hop attacks—the MIAs designed for i.i.d. data and adapted to graph data by ignoring the edges. This is not surprising since G-BASE uses a more complex loss signal and has to estimate an expectation over target node neighborhoods. However, it is the inability to parallelize the membership inference over multiple target nodes that is the true bottleneck of G-BASE in our experimental settings. Since the loss signal of G-BASE includes a term that depends on the difference in loss values over a neighborhood when the target node is included and when it is not (see Figure 2), it is not trivial to parallelize the computation of the loss signals over several target nodes. The 0-hop MIAs, on the other hand, can compute their loss signals over a large batch of target nodes in parallel. Thus, the time complexity of inference with G-BASE has an additional factor of $N \times M$ compared to BASE, where $N$ is the number of target nodes and $M$ is the number of sampled graphs. We found that G-BASE performs well with only $M = 8$ (see Appendix H.2 for results regarding this), but the number of target nodes is typically on the order of thousands, and consequently this factor is the dominating factor. Nevertheless, G-BASE remains computationally tractable, and we believe it is ultimately more important to accurately assess data leakage than to be efficient, in agreement with [11]. However, further improvements in the efficiency and parallelization of the attack are interesting and important directions of future research. In Appendix H.3, we quantify the efficiency of our attacks, LiRA, and RMIA in terms of wall-clock time.

**Conclusions.** We proposed BASE and G-BASE, practical and theoretically-grounded MIAs for i.i.d. and graph data. By deriving the Bayes-optimal inference rule for node-level attacks on GNNs, we addressed key challenges posed by structural dependencies in graphs. Our attacks match or surpass existing state-of-the-art methods (LiRA and RMIA) while, in the case of BASE, requiring significantly lower computational overhead. Our results bridge the gap between theoretical optimality and practical implementation of MIAs, offering an efficient framework for privacy auditing across both classical and graph-based learning settings.

# 7 Acknowledgments and Disclosure of Funding

This work was partially supported by the Wallenberg AI, Autonomous Systems and Software Program (WASP) funded by the Knut and Alice Wallenberg Foundation, by the Swedish Research Council (VR) under grants 2020-03687 and 2023-05065, and by Vinnova under grant 2023-03000.

The computations were enabled by resources provided by the National Academic Infrastructure for Supercomputing in Sweden (NAISS), partially funded by the Swedish Research Council through grant agreement no. 2022-06725.

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

# A Experimental Setup

**Target Model Training.** We consider target models trained in a supervised manner for node classification tasks, using the commonly adopted cross-entropy loss as the objective function. We evaluate three target model architectures: graph convolutional networks (GCNs) [30], GraphSAGE [31] with max aggregation, and graph attention networks (GATs) [32] using 4 attention heads in the first layer and 2 in the second. Optimization is performed using Adam [35]. For each dataset and model, hyperparameters are selected via a grid search, including the learning rate, weight decay, number of training epochs, dropout rate, and dimension of the first GNN layer.[3] In particular, we search over $\{0.01, 0.001\}$ for the learning rate, $\{0.0001, 0.00001\}$ for the weight decay, and $\{0.0, 0.25, 0.5\}$ for the dropout rate. For the hidden dimension of the first layer, we search in $\{32, 64, 128, 256, 512\}$, with 32 or 512 excluded depending on the dataset. The initial search space for the number of epochs is typically $\{20, 50, 100, 200, 400, 800, 1600\}$, and is sometimes later refined. Target models with very small generalization gaps are often difficult to attack, making it harder to discern differences in MIA performance. To avoid this issue, we restrict the grid search to configurations that yield an average generalization gap of at least 8%. This represents a realistic generalization gap, sufficient to enable meaningful attacks without excessive overfitting that would distort expected attack performance. We emphasize that our goal is not to produce the best performing target model, but rather to obtain well-performing target models with a representative generalization gap.

The dataset-model combinations evaluated in Table 1 correspond to the best-performing model architecture selected for each dataset. The corresponding train and test accuracies of the target models are reported in Table 2.

Table 2: Train and test accuracy of the target model on the different datasets and architectures used in Table 1. The accuracies are reported as mean $\pm$ standard deviation, over the 10 different target models used in all except the mismatched adversary experiments.

| Dataset (model) | Train accuracy (%) | Test accuracy (%) |
|---|---|---|
| Cora (GCN) | $96.07 \pm 0.37$ | $80.67 \pm 1.10$ |
| Citeseer (GAT) | $92.36 \pm 0.43$ | $73.80 \pm 0.97$ |
| Pubmed (GraphSAGE) | $96.30 \pm 0.26$ | $87.14 \pm 0.28$ |
| Flickr (GCN) | $57.48 \pm 1.05$ | $47.08 \pm 0.61$ |
| Amazon-photo (GAT) | $99.77 \pm 0.06$ | $91.44 \pm 0.77$ |
| Github (GraphSAGE) | $93.68 \pm 1.88$ | $84.37 \pm 0.91$ |

**Shadow model training.** Shadow models are trained using the same hyperparameter settings as the target model, implicitly assuming adversarial side-knowledge. This setting is particularly relevant for MIA auditing, as it yields an upper bound on the attack performance. To facilitate efficient MIA auditing in the online setting (see Appendix F for a discussion of online vs. offline settings), we adopt the shadow model training procedure proposed in [11] and also used in [13]. Specifically, each shadow model is trained on half of the data population (e.g., half the nodes in a graph dataset), such that each data sample is included in the training set of half of the models. Pseudo-code for our precise shadow model training procedure is provided in Algorithm 1. For graph data, the data population is a graph dataset, and sampling data points corresponds to sampling nodes, retaining the edges between sampled nodes. This procedure guarantees a balanced set of in-models (models trained with the target data point) and out-models (models trained without it) for each data point. In the offline setting, for each target sample, the in-models for that sample are filtered out, so that only out-models are used in the attack. This filtering approach eliminates the need for a separate, disjoint dataset to train shadow models, an important advantage in graph-data settings, since splitting a graph in disjoint parts reduces the number of edges, leading to sparse graphs when the full graph has low average degree. The downside of the filtering approach is that only half of the shadow models are used for attacking any given target sample.

---

[3]The second layer always produces embeddings with dimensionality equal to the number of classes.

**Algorithm 1** Shadow Model Training Procedure.

---

1: **Input:** Data population $\mathcal{G}$, training algorithm $\mathcal{T}$, and even number of shadow models $2N$.
2: $\Phi \leftarrow \emptyset$
3: **for** $k = 1$ to $N$ **do**
4: $\quad \mathcal{G}_k \sim \text{Uniform}(\mathcal{G})$, $|\mathcal{G}_k| = \frac{1}{2}|\mathcal{G}|$
5: $\quad \mathcal{G}_k^c = \{z : z \in \mathcal{G}, z \notin \mathcal{G}_k\}$
6: $\quad \phi_k \leftarrow \mathcal{T}(\mathcal{G}_k)$
7: $\quad \phi_k^c \leftarrow \mathcal{T}(\mathcal{G}_k^c)$
8: $\quad \Phi \leftarrow \Phi \cup \{\phi_k, \phi_k^c\}$
9: **end for**
10: **return** $\Phi$

---

## B  Bayes-Optimal Node-Level MIAs Against GNNs

### B.1  Proof of Theorem 1

The proof follows along the lines of the proofs of [1, Thms. 1 and 2] for the case of i.i.d. data. We begin by applying the law of total expectation to express $P(m_v = 1|\boldsymbol{\theta}, \mathcal{G})$ as an expectation over the unknown membership statuses of the remaining nodes,

$$P(m_v = 1|\boldsymbol{\theta}, \mathcal{G}) = \mathbb{E}_{\tilde{\mathcal{M}} \sim P(\tilde{\mathcal{M}}|\boldsymbol{\theta}, \mathcal{G})}[P(m_v = 1|\tilde{\mathcal{M}}, \boldsymbol{\theta}, \mathcal{G})]. \tag{11}$$

Unlike [1], where the expectation is taken over the data samples, our threat model assumes that the full graph $\mathcal{G}$ is known to the adversary. Consequently, we marginalize only over the unknown membership indicator variables of the non-target nodes.

The term $P(m_v = 1|\tilde{\mathcal{M}}, \boldsymbol{\theta}, \mathcal{G})$ can be computed using Bayes' rule, yielding

$$P(m_v = 1|\boldsymbol{\theta}, \mathcal{G}) = \mathbb{E}_{\tilde{\mathcal{M}} \sim P(\tilde{\mathcal{M}}|\boldsymbol{\theta}, \mathcal{G})}\left[\sigma\left(\log \frac{p(\boldsymbol{\theta}|m_v = 1, \tilde{\mathcal{M}}, \mathcal{G})}{p(\boldsymbol{\theta}|m_v = 0, \tilde{\mathcal{M}}, \mathcal{G})} + \log \frac{\lambda}{1 - \lambda}\right)\right], \tag{12}$$

where $\lambda = P(m_v = 1|\tilde{\mathcal{M}}, \mathcal{G})$ is the prior probability that node $v$ is a member of the training set, before observing the model. Under our threat model, conditioning only on the data and the membership indicators of other nodes does not provide any information about the target node membership status. Hence, $\lambda = P(m_v = 1)$.

Finally, we need to compute the log ratio of model posteriors in (12). Assuming the negative log-likelihood loss function defined in (3), and making the natural assumption that, given $\mathcal{M}$, the model depends only on the subgraph of $\mathcal{G}$ induced by $\mathcal{M}$—that is, it is influenced solely by the data it was trained on—the posterior distribution of the model parameters can be expressed using Bayes' rule in terms of the loss function and a prior $p(\boldsymbol{\theta}|\mathcal{M}, \boldsymbol{X}, \boldsymbol{A})$ as

$$\begin{aligned} p(\boldsymbol{\theta}|\mathcal{M}, \mathcal{G}) &= \frac{p(\{\boldsymbol{y}_v\}_{v:m_v=1}|\boldsymbol{\theta}, \mathcal{M}, \boldsymbol{X}, \boldsymbol{A})p(\boldsymbol{\theta}|\mathcal{M}, \boldsymbol{X}, \boldsymbol{A})}{\int p(\{\boldsymbol{y}_v\}_{v:m_v=1}|\boldsymbol{\phi}, \mathcal{M}, \boldsymbol{X}, \boldsymbol{A})p(\boldsymbol{\phi}|\mathcal{M}, \boldsymbol{X}, \boldsymbol{A})\mathrm{d}\boldsymbol{\phi}} \\ &\overset{(a)}{=} \frac{\prod_{v:m_v=1} p(\boldsymbol{y}_v|\boldsymbol{\theta}, \mathcal{M}, \boldsymbol{X}, \boldsymbol{A})p(\boldsymbol{\theta}|\mathcal{M}, \boldsymbol{X}, \boldsymbol{A})}{\int \prod_{v:m_v=1} p(\boldsymbol{y}_v|\boldsymbol{\phi}, \mathcal{M}, \boldsymbol{X}, \boldsymbol{A})p(\boldsymbol{\phi}|\mathcal{M}, \boldsymbol{X}, \boldsymbol{A})\mathrm{d}\boldsymbol{\phi}} \\ &\overset{(b)}{=} \frac{e^{-\sum_{v \in \mathcal{V}} m_v \ell(f_{\boldsymbol{\theta}}(\boldsymbol{X}, \boldsymbol{A}_{\mathcal{M}})_v, \boldsymbol{y}_v)}p(\boldsymbol{\theta}|\mathcal{M}, \boldsymbol{X}, \boldsymbol{A})}{\int e^{-\sum_{v \in \mathcal{V}} m_v \ell(f_{\boldsymbol{\phi}}(\boldsymbol{X}, \boldsymbol{A}_{\mathcal{M}})_v, \boldsymbol{y}_v)}p(\boldsymbol{\phi}|\mathcal{M}, \boldsymbol{X}, \boldsymbol{A})\mathrm{d}\boldsymbol{\phi}}, \end{aligned} \tag{13}$$

where $(a)$ follows since, from (1), each embedding $\boldsymbol{z}_v$ is a function of $(\boldsymbol{X}, \boldsymbol{A}, \boldsymbol{\theta}, \mathcal{M})$; by [36, Sec. 3.3], this implies that $\boldsymbol{z}_v$ is independent of $\boldsymbol{z}_u$ given $(\boldsymbol{X}, \boldsymbol{A}, \boldsymbol{\theta}, \mathcal{M})$ for all $v \neq u$. Applying the softmax and indexing with $\boldsymbol{y}_v$, see (2), preserves this independence. Step $(b)$ follows from the definition of the loss function in (3). Also, the $n \times n$ matrices $\boldsymbol{A}_{\mathcal{M}}$ and $\boldsymbol{A}_{\tilde{\mathcal{M}}}$ are defined as

$$(\boldsymbol{A}_{\mathcal{M}})_{uw} = \begin{cases} A_{uw}, & m_u = m_w = 1, \\ 0, & \text{otherwise} \end{cases} \quad \text{and} \quad (\boldsymbol{A}_{\tilde{\mathcal{M}}})_{uw} = \begin{cases} (\boldsymbol{A}_{\mathcal{M}})_{uw}, & u, w \neq v, \\ 0, & \text{otherwise} \end{cases}$$

for $u, w \in \mathcal{V}$. In words, $\boldsymbol{A}_{\mathcal{M}}$ is the adjacency matrix of the subgraph of $\mathcal{G}$ induced by the nodes masked by $\mathcal{M}$, while $\boldsymbol{A}_{\tilde{\mathcal{M}}}$ corresponds to the subgraph of this resulting graph obtained by removing the target node and all its adjacent edges.

We can now evaluate the log likelihood-ratio in terms of the loss function:

$$\log \frac{p(\boldsymbol{\theta}|m_v = 1, \tilde{\mathcal{M}}, \mathcal{G})}{p(\boldsymbol{\theta}|m_v = 0, \tilde{\mathcal{M}}, \mathcal{G})}$$

$$= -\sum_{u \in \mathcal{V}} m_u \ell(f_\theta(\boldsymbol{X}, \boldsymbol{A}_\mathcal{M})_u, \boldsymbol{y}_u) + \sum_{u \in \mathcal{V} \setminus \{v\}} m_u \ell(f_\theta(\boldsymbol{X}, \boldsymbol{A}_{\tilde{\mathcal{M}}})_u, \boldsymbol{y}_u)$$

$$+ \log \frac{p(\boldsymbol{\theta}|m_v = 1, \tilde{\mathcal{M}}, \boldsymbol{X}, \boldsymbol{A})}{p(\boldsymbol{\theta}|m_v = 0, \tilde{\mathcal{M}}, \boldsymbol{X}, \boldsymbol{A})}$$

$$- \log \frac{\int e^{-\sum_{u \in \mathcal{V}} m_u \ell(f_\phi(\boldsymbol{X}, \boldsymbol{A}_\mathcal{M})_u, \boldsymbol{y}_u)} p(\boldsymbol{\phi}|m_v = 1, \tilde{\mathcal{M}}, \boldsymbol{X}, \boldsymbol{A}) \mathrm{d}\boldsymbol{\phi}}{\int e^{-\sum_{u \in \mathcal{V} \setminus \{v\}} m_u \ell(f_{\phi'}(\boldsymbol{X}, \boldsymbol{A}_{\tilde{\mathcal{M}}})_u, \boldsymbol{y}_u)} p(\boldsymbol{\phi}'|m_v = 0, \tilde{\mathcal{M}}, \boldsymbol{X}, \boldsymbol{A}) \mathrm{d}\boldsymbol{\phi}'}$$

$$\overset{(c)}{=} -\ell(f_{\boldsymbol{\theta}}(\boldsymbol{X}, \boldsymbol{A}_\mathcal{M})_v, \boldsymbol{y}_v) - \Delta\mathcal{L}_{\mathcal{N}_L(v)} + \log \frac{p(\boldsymbol{\theta}|m_v = 1, \tilde{\mathcal{M}}, \boldsymbol{X}, \boldsymbol{A})}{p(\boldsymbol{\theta}|m_v = 0, \tilde{\mathcal{M}}, \boldsymbol{X}, \boldsymbol{A})}$$

$$- \log \int e^{-\ell(f_\phi(\boldsymbol{X}, \boldsymbol{A}_\mathcal{M})_v, \boldsymbol{y}_v) - \Delta\mathcal{L}_{\mathcal{N}_L(v)}} \frac{p(\boldsymbol{\phi}|m_v = 1\tilde{\mathcal{M}}, \boldsymbol{X}, \boldsymbol{A})}{p(\boldsymbol{\phi}|m_v = 0\tilde{\mathcal{M}}, \boldsymbol{X}, \boldsymbol{A})}$$

$$\times \frac{e^{-\sum_{u \in \mathcal{V} \setminus \{v\}} m_u \ell(f_\phi(\boldsymbol{X}, \boldsymbol{A}_{\tilde{\mathcal{M}}})_u, \boldsymbol{y}_u)} p(\boldsymbol{\phi}|m_v = 0, \tilde{\mathcal{M}}, \boldsymbol{X}, \boldsymbol{A})}{\int e^{-\sum_{u \in \mathcal{V} \setminus \{v\}} m_u \ell(f_{\phi'}(\boldsymbol{X}, \boldsymbol{A}_{\tilde{\mathcal{M}}})_u, \boldsymbol{y}_u)} p(\boldsymbol{\phi}'|m_v = 0, \tilde{\mathcal{M}}, \boldsymbol{X}, \boldsymbol{A}) \mathrm{d}\boldsymbol{\phi}'} \mathrm{d}\boldsymbol{\phi}$$

$$\overset{(d)}{=} -\ell(f_{\boldsymbol{\theta}}(\boldsymbol{X}, \boldsymbol{A}_\mathcal{M})_v, \boldsymbol{y}_v) - \Delta\mathcal{L}_{\mathcal{N}_L(v)} + \log \Lambda(\boldsymbol{\theta}, \tilde{\mathcal{M}}, \boldsymbol{X}, \boldsymbol{A})$$

$$- \log \int e^{-\ell(f_\phi(\boldsymbol{X}, \boldsymbol{A}_\mathcal{M})_v, y_v) - \Delta\mathcal{L}_{\mathcal{N}_L(v)} + \log \Lambda(\phi, \tilde{\mathcal{M}}, \boldsymbol{X}, \boldsymbol{A})} p(\boldsymbol{\phi}|m_v = 0, \tilde{\mathcal{M}}, \mathcal{G}) \mathrm{d}\boldsymbol{\phi}, \qquad (14)$$

where in step $(c)$ we have used

$$- \sum_{u \in \mathcal{V}} m_u \ell(f_\theta(\boldsymbol{X}, \boldsymbol{A}_\mathcal{M})_u, \boldsymbol{y}_u) + \sum_{u \in \mathcal{V} \setminus \{v\}} m_u \ell(f_\theta(\boldsymbol{X}, \boldsymbol{A}_{\tilde{\mathcal{M}}})_u, \boldsymbol{y}_u)$$

$$= -\ell(f_{\boldsymbol{\theta}}(\boldsymbol{X}, \boldsymbol{A}_\mathcal{M})_v, \boldsymbol{y}_v) - \sum_{u \in \mathcal{V} \setminus \{v\}} m_u(\ell(f_\theta(\boldsymbol{X}, \boldsymbol{A}_\mathcal{M})_u, \boldsymbol{y}_u) - \ell(f_\theta(\boldsymbol{X}, \boldsymbol{A}_{\tilde{\mathcal{M}}})_u, \boldsymbol{y}_u))$$

$$= -\ell(f_{\boldsymbol{\theta}}(\boldsymbol{X}, \boldsymbol{A}_\mathcal{M})_v, \boldsymbol{y}_v) - \sum_{u \in \mathcal{N}_L(v)} m_u(\ell(f_\theta(\boldsymbol{X}, \boldsymbol{A}_\mathcal{M})_u, \boldsymbol{y}_u) - \ell(f_\theta(\boldsymbol{X}, \boldsymbol{A}_{\tilde{\mathcal{M}}})_u, \boldsymbol{y}_u))$$

$$= -\ell(f_{\boldsymbol{\theta}}(\boldsymbol{X}, \boldsymbol{A}_\mathcal{M})_v, \boldsymbol{y}_v) - \Delta\mathcal{L}_{\mathcal{N}_L(v)}$$

since the inclusion or exclusion of node $v$ affects only the predictions within its $L$-hop neighborhood $\mathcal{N}_L(v)$, assuming an $L$-layer GNN. Furthermore, in step $(d)$ we used (13) to simplify the model distribution that is integrated over and substituted $\Lambda(\boldsymbol{\phi}, \tilde{\mathcal{M}}, \boldsymbol{X}, \boldsymbol{A}) = \frac{p(\boldsymbol{\phi}|m_v = 1, \tilde{\mathcal{M}}, \boldsymbol{X}, \boldsymbol{A})}{p(\boldsymbol{\phi}|m_v = 0, \tilde{\mathcal{M}}, \boldsymbol{X}, \boldsymbol{A})}$ for the prior likelihood ratio (before observing the labels).

Combining (12) and (14) concludes the proof.

## B.2  G-BASE: Practical Bayes-Optimal MIA against GNNs

We elaborate on the proposed sampling strategies used to approximate the expectation over the membership statuses of all non-target samples, $\tilde{\mathcal{M}}$, i.e., the term $\mathbb{E}_{\tilde{\mathcal{M}} \sim P(\tilde{\mathcal{M}}|\boldsymbol{\theta}, \mathcal{G})}$ in Equation (5).

In practice, we compute samples of all the indicator variables $\mathcal{M}$ and obtain $\tilde{\mathcal{M}}$ by discarding the indicator variable of the target node. This allows us to get samples of $\tilde{\mathcal{M}}$ for multiple target nodes from a single sample $\mathcal{M}$.

**Model-independent sampling.** The simplest approximation assumes that the membership inference configuration $\mathcal{M}$ is independent of the target model, thereby ignoring the conditioning on $\boldsymbol{\theta}$, i.e., $P(\mathcal{M}|\boldsymbol{\theta}, \mathcal{G}) \approx P(\mathcal{M}|\mathcal{G})$. Under this assumption, we approximate $P(\mathcal{M}|\mathcal{G})$ by treating each membership indicator $m_v \in \mathcal{M}$ as i.i.d. according to a Bernoulli distribution with parameter $\lambda = P(m_v = 1)$. We then generate samples of $\mathcal{M}$ as $\mathcal{M} = \{m_v : m_v \sim \text{Ber}(\lambda), v \in \mathcal{V}\}$.

$0$**-hop MIA sampling.** Recall that $P(\mathcal{M}|\boldsymbol{\theta}, \mathcal{G})$ is the joint distribution of the membership of all nodes. A natural approximation is to apply a per-node MIA to estimate individual membership probabilities

and assume independence across nodes. Concretely, we apply a $0$-hop MIA—i.e., we ignore the graph structure and treat nodes as i.i.d. samples. This reduces the problem to the standard i.i.d. setting, where well-established MIA methods can be applied. Any i.i.d.-based MIA that yields membership scores convertible to probabilities can be used. Here, we adopt the attack introduced in Section 5.2 which approximates the Bayes-optimal inference rule in the i.i.d setting. We then generate samples from the approximate posterior as $\mathcal{M} = \{m_v : m_v \sim \text{Ber}(P(m_v = 1|\boldsymbol{\theta}, \boldsymbol{X}_v, \boldsymbol{y}_v)), v \in \mathcal{V}\}$, where $P(m_v = 1|\boldsymbol{\theta}, \boldsymbol{X}_v, \boldsymbol{y}_v)$ is the membership probability of node $v$ assigned by the BASE attack.

**Metropolis-Hastings sampling.** To account for the dependence of $\mathcal{M}$ on the target model $\boldsymbol{\theta}$, we develop a Markov chain Monte Carlo (MCMC) method based on the Metropolis-Hastings algorithm. The goal is to construct a Markov chain over membership configurations $\mathcal{M}$ with $P(\mathcal{M}|\boldsymbol{\theta}, \mathcal{G})$ as its stationary distribution. To apply Metropolis-Hastings, we need to be able to evaluate the unnormalized (up to a multiplicative constant) probability mass function $P(\mathcal{M}|\boldsymbol{\theta}, \mathcal{G})$, i.e., a function $P^*(\mathcal{M}|\boldsymbol{\theta}, \mathcal{G})$ satisfying

$$P(\mathcal{M}|\boldsymbol{\theta}, \mathcal{G}) = \frac{P^*(\mathcal{M}|\boldsymbol{\theta}, \mathcal{G})}{\sum_{\mathcal{M}'} P^*(\mathcal{M}'|\boldsymbol{\theta}, \mathcal{G})}.$$

We derive such a function using Bayes' rule, assuming a uniform prior over $\mathcal{M}$ to eliminate the prior terms,

$$P(\mathcal{M}|\boldsymbol{\theta}, \mathcal{G}) = \frac{p(\boldsymbol{\theta}|\mathcal{M}, \mathcal{G})}{\sum_{\mathcal{M}'} p(\boldsymbol{\theta}|\mathcal{M}', \mathcal{G})}.$$

Using (13) for the model posterior and assuming that the prior (before observing the labels) is independent of $\mathcal{M}$ and only depends on the graph under consideration, we obtain

$$P(\mathcal{M}|\boldsymbol{\theta}, \mathcal{G}) \propto \frac{e^{-\sum_{u \in \mathcal{V}} m_u \ell(f_{\boldsymbol{\theta}}(\boldsymbol{X}, \boldsymbol{A}_{\mathcal{M}})_u, \boldsymbol{y}_u)}}{\int e^{-\sum_{u \in \mathcal{V}} m_u \ell(f_{\boldsymbol{\phi}}(\boldsymbol{X}, \boldsymbol{A}_{\mathcal{M}})_u, \boldsymbol{y}_u)} p(\boldsymbol{\phi}) \mathrm{d}\boldsymbol{\phi}} = P^*(\mathcal{M}|\boldsymbol{\theta}, \mathcal{G}). \tag{15}$$

Our proposal distribution $Q(\mathcal{M}'|\mathcal{M})$ randomly flips a small proportion $\epsilon$ of the indicator variables in the current state $\mathcal{M}$. Notably, this proposal distribution is symmetric, $Q(\mathcal{M}'|\mathcal{M}) = Q(\mathcal{M}|\mathcal{M}')$, so it cancels from the Metropolis-Hastings acceptance ratio. To sample from this distribution, we initialize the Markov chain at a randomly-selected configuration $\mathcal{M}^{(0)}$. At iteration $t$, we propose a new configuration $\mathcal{M}^*$ by flipping a fraction $\epsilon$ of the membership indicators in $\mathcal{M}^{(t)}$, and compute the log acceptance ratio

$$\begin{aligned}
\log \frac{p(\boldsymbol{\theta}|\mathcal{M}^*, \mathcal{G})}{p(\boldsymbol{\theta}|\mathcal{M}^{(t)}, \mathcal{G})} &= \sum_{u \in \mathcal{V}} (m_u^{(t)} \ell(f_{\boldsymbol{\theta}}(\boldsymbol{X}, \boldsymbol{A}_{\mathcal{M}^{(t)}})_u, \boldsymbol{y}_u) - m_u^* \ell(f_{\boldsymbol{\theta}}(\boldsymbol{X}, \boldsymbol{A}_{\mathcal{M}^*})_u, \boldsymbol{y}_u)) \\
&+ \log \left( \int e^{-\sum_{u \in \mathcal{V}} m_u^{(t)} \ell(f_{\boldsymbol{\theta}}(\boldsymbol{X}, \boldsymbol{A}_{\mathcal{M}^{(t)}})_u, \boldsymbol{y}_u)} p(\boldsymbol{\phi}) \mathrm{d}\boldsymbol{\phi} \right) \\
&- \log \left( \int e^{-\sum_{u \in \mathcal{V}} m_u^* \ell(f_{\boldsymbol{\theta}}(\boldsymbol{X}, \boldsymbol{A}_{\mathcal{M}^*})_u, \boldsymbol{y}_u)} p(\boldsymbol{\phi}) \mathrm{d}\boldsymbol{\phi} \right).
\end{aligned} \tag{16}$$

Each integral can be efficiently approximated via Monte Carlo sampling using shadow models. Notably, the same shadow models used to approximate the inner expectation in (5) can be reused here to evaluate (16). We then accept the proposal with probability $\min(1, p(\boldsymbol{\theta}|\mathcal{M}^*, \mathcal{G})/p(\boldsymbol{\theta}|\mathcal{M}^{(t)}, \mathcal{G}))$. Specifically, we draw $u \sim \text{Uniform}(0, 1)$ and set

$$(\mathcal{M}^{(t+1)}, p(\boldsymbol{\theta}|\mathcal{M}^{(t+1)}, \mathcal{G})) = \begin{cases} (\mathcal{M}^*, p(\boldsymbol{\theta}|\mathcal{M}^*, \mathcal{G})) & \text{if } \frac{p^*}{p^{(t)}} > u \\ (\mathcal{M}^{(t)}, p(\boldsymbol{\theta}|\mathcal{M}^{(t)}, \mathcal{G})) & \text{otherwise.} \end{cases}$$

To obtain approximately independent samples, we insert a burn-in period at the beginning of the chain and use thinning—collecting samples only after a sufficient number of iterations.

**Gibbs sampling.** Another MCMC sampling strategy that is well suited to our setup is Gibbs sampling. The sample a an instance of indicator variables $\mathcal{M} \sim P(\mathcal{M}|\boldsymbol{\theta}, \mathcal{M})$, the Gibbs method iteratively samples individual components conditioned on the remaining indicator variables. More precisely, the $k$th iteration of Gibbs sampling produces a sample $\mathcal{M}^{(k)} = \{m_1^{(k)}, \ldots, m_n^{(k)}\}$ by sampling the

components sequentially:

$$m_1^{(k)} \sim P(m_1|m_2^{(k-1)}, \ldots, m_n^{(k-1)}, \boldsymbol{\theta}, \mathcal{G})$$
$$m_2^{(k)} \sim P(m_2|m_1^{(k)}, m_3^{(k-1)}, \ldots, m_n^{(k-1)}, \boldsymbol{\theta}, \mathcal{G})$$
$$\vdots$$
$$m_n^{(k)} \sim P(m_n|m_1^{(k)}, \ldots, m_{n-1}^{(k)}, \boldsymbol{\theta}, \mathcal{G})$$

We note that each of these conditional probabilities is given by the Bayes-optimal membership inference formula (5):

$$P(m_v = 1|\tilde{\mathcal{M}}, \boldsymbol{\theta}, \mathcal{G})$$
$$= \sigma\left(-S_{\mathcal{L}}(f_{\boldsymbol{\theta}}, v, \tilde{\mathcal{M}}, \mathcal{G}) - \log \int e^{-S_{\mathcal{L}}(f_{\boldsymbol{\phi}}, v, \tilde{\mathcal{M}}, \mathcal{G})} p(\boldsymbol{\phi}|m_v = 0, \tilde{\mathcal{M}}, \mathcal{G})\mathrm{d}\boldsymbol{\phi} + \log \frac{\lambda}{1-\lambda}\right).$$

Sampling from the conditional distributions simply amounts to sample from a Bernoulli distribution:

$$m_u \sim \mathrm{Ber}(P(m_u|m_1, \ldots, m_{u-1}, m_{u+1}, \ldots, m_n, \boldsymbol{\theta}, \mathcal{G})).$$

This Gibbs sampling strategy is therefore a kind of MIA-based sampling that improves on the 0-hop MIA sampling by accounting for the graph structure. A further advantage of the Gibbs sampling strategy over the Metropolis-Hastings method is that it does not require tuning any parameters. However, it is computationally demanding since a single iteration requires querying the models over each node in the full graph $\mathcal{G}$.

## C  Selecting the Decision Threshold

Since an adversary does not have access to ground truth membership labels, they cannot directly tune the decision threshold (by sweeping $\tau$) to achieve a specific FPR. Instead, the adversary must choose a threshold that is expected to yield an FPR close to the maximal tolerated FPR. For MIAs based on shadow models, we propose finding such a threshold by designating a subset of the shadow models as simulated target models, as suggested in [1, 37]. The simulated target models can be attacked using the remaining shadow models, possibly employing cross-validation. Because the ground-truth training data is known for the shadow models, the adversary can sweep over decision thresholds and identify the values that are expected to approximately yield the desired FPR.

By repeating the threshold estimation process across multiple simulated target models, the adversary can assess the variability in the resulting thresholds. A conservative adversary would choose a threshold at least as large as the maximum threshold obtained. Another viable option is to choose the mean of the thresholds. As shown in Table 3 (graph data) and Table 4 (i.i.d. data), the variation in threshold values across target models is small, indicating that thresholds estimated from simulated target models are fairly stable. RMIA exhibits the lowest threshold variance. However, the viability of estimating the threshold using simulated target models also depends on how sensitive the resulting FPR is to threshold fluctuations. Therefore, to demonstrate the effectiveness of this approach, we train 10 shadow models to act as simulated target models. These simulated target models are attacked and, using the ground truth knowledge about the training members, the decision thresholds resulting in an FPR not exceeding 1% are computed. The estimated threshold is then taken as the average of these decision thresholds. To run the attacks, 8 (online) and 4 (offline) separate shadow models are used. Table 5 shows the TPR and FPR achieved when using the estimated threshold on real target models. The FPR obtained using the estimated threshold does not deviate much from the target 1% FPR. Hence, the TPR is also close to the TPR of the exact 1% FPR threshold. Despite the differences in threshold variance between LiRA, RMIA, and our attacks (see Table 3 and Table 4), there is no significant difference in the accuracy of the estimated threshold.

We conclude with a remark on alternative methods for threshold selection. For LiRA, the threshold can be selected from the fitted Gaussian distribution. However, the accuracy of the estimated threshold depends on how well the Gaussian distribution fits the logit-scaled confidence values. As such, this approach also relies on an estimation based on population data, and on top, the heuristic observation that logit-scaled confidence values often look normal distributed. For RMIA with $\gamma = 1$, the authors argue that their attack is calibrated such that a threshold $\beta = 1 - \alpha$ results in FPR $\alpha$. Investigating

Table 3: Decision thresholds for RMIA, BASE and G-BASE resulting in the largest FPR less than or equal to the target FPR (1% and 0.1%). The threshold is reported as the mean ± standard deviation, averaged over 10 different target models and sets of target samples. A comparatively small standard deviation indicates that the threshold is not expected to vary too much over different target models, allowing the adversary to estimate a threshold using shadow models as simulated target models. The RMIA threshold does not satisfiy the empirical rule "threshold = 1 − FPR" as reported in [13].

| K | ATTACK | CORA (GCN) | | CITESEER (GAT) | | PUBMED (GraphSAGE) | |
| | | THRESHOLD@FPR | | THRESHOLD@FPR | | TPR@FPR | |
| | | 1% | 0.1% | 1% | 0.1% | 1% | 0.1% |
|---|---|---|---|---|---|---|---|
| 8 | RMIA | $0.8566 \pm 0.0078$ | $0.9030 \pm 0.0174$ | $0.8240 \pm 0.0124$ | $0.8990 \pm 0.0299$ | $0.9474 \pm 0.0039$ | $0.9822 \pm 0.0030$ |
| | **BASE** | $0.6024 \pm 0.0098$ | $0.6913 \pm 0.0342$ | $0.5907 \pm 0.0097$ | $0.6971 \pm 0.0548$ | $0.6052 \pm 0.0093$ | $0.7140 \pm 0.0161$ |
| | **G-BASE** (MIA) | $0.5619 \pm 0.0072$ | $0.6204 \pm 0.0327$ | $0.5724 \pm 0.0059$ | $0.6471 \pm 0.0309$ | $0.5928 \pm 0.0053$ | $0.6771 \pm 0.0175$ |
| 4 | RMIA (OFF) | $0.8275 \pm 0.0087$ | $0.8612 \pm 0.0128$ | $0.7649 \pm 0.0101$ | $0.8315 \pm 0.0145$ | $0.9719 \pm 0.0027$ | $0.9926 \pm 0.0016$ |
| | **BASE** (OFF) | $0.5358 \pm 0.0062$ | $0.6015 \pm 0.0454$ | $0.5540 \pm 0.0075$ | $0.6341 \pm 0.0368$ | $0.6240 \pm 0.0093$ | $0.7422 \pm 0.0182$ |
| | **G-BASE** (OFF) | $0.7034 \pm 0.0127$ | $0.8160 \pm 0.0324$ | $0.7064 \pm 0.0168$ | $0.8466 \pm 0.0311$ | $0.6661 \pm 0.0053$ | $0.8031 \pm 0.0213$ |

| K | ATTACK | FLICKR (GCN) | | AMAZON-PHOTO (GAT) | | GITHUB (GraphSAGE) | |
| | | THRESHOLD@FPR | | THRESHOLD@FPR | | THRESHOLD@FPR | |
| | | 1% | 0.1% | 1% | 0.1% | 1% | 0.1% |
|---|---|---|---|---|---|---|---|
| 8 | RMIA | $0.8029 \pm 0.0086$ | $0.9142 \pm 0.0099$ | $0.9267 \pm 0.0101$ | $0.9675 \pm 0.0144$ | $0.9462 \pm 0.0053$ | $0.9785 \pm 0.0038$ |
| | **BASE** | $0.9080 \pm 0.0091$ | $0.9873 \pm 0.0036$ | $0.7167 \pm 0.0392$ | $0.8866 \pm 0.0762$ | $0.6328 \pm 0.0178$ | $0.7250 \pm 0.0300$ |
| | **G-BASE** (MIA) | $0.6770 \pm 0.0108$ | $0.7910 \pm 0.0109$ | $0.6031 \pm 0.0188$ | $0.7316 \pm 0.0532$ | $0.6162 \pm 0.0114$ | $0.6990 \pm 0.0189$ |
| 4 | RMIA (OFF) | $0.8695 \pm 0.0074$ | $0.9330 \pm 0.0042$ | $0.8679 \pm 0.0084$ | $0.9032 \pm 0.0055$ | $0.9490 \pm 0.0119$ | $0.9801 \pm 0.0056$ |
| | **BASE** (OFF) | $0.9143 \pm 0.0084$ | $0.9884 \pm 0.0036$ | $0.5806 \pm 0.0119$ | $0.7219 \pm 0.0830$ | $0.6317 \pm 0.0233$ | $0.7421 \pm 0.0258$ |
| | **G-BASE** (OFF) | $0.6965 \pm 0.0109$ | $0.8072 \pm 0.0191$ | $0.6822 \pm 0.0121$ | $0.8604 \pm 0.0279$ | $0.6677 \pm 0.0108$ | $0.7826 \pm 0.0184$ |

Table 4: Decision thresholds for BASE (without the sigmoid normalization), RMIA and LiRA resulting in the largest FPR less than or equal to the target FPR (1% and 0.1%). The threshold is reported as the mean ± standard deviation, averaged over 10 different target models. A comparatively small standard deviation indicates that the threshold is not expected to vary too much over different target models, allowing the adversary to estimate a threshold using shadow models as simulated target models. The RMIA threshold does not satisfy the empirical rule threshold=1-FPR as reported in [13].

| K | ATTACK | CIFAR-10 | | CIFAR-100 | |
| | | THRESHOLD@FPR | | THRESHOLD@FPR | |
| | | 1% | 0.1% | 1% | 0.1% |
|---|---|---|---|---|---|
| 32 | **BASE** | $0.5385 \pm 0.0473$ | $0.8159 \pm 0.0669$ | $0.8417 \pm 0.0735$ | $1.2240 \pm 0.0914$ |
| | RMIA | $0.9624 \pm 0.0047$ | $0.9900 \pm 0.0028$ | $0.9429 \pm 0.0139$ | $0.9833 \pm 0.0061$ |
| | LiRA | $0.6505 \pm 0.0515$ | $1.1098 \pm 0.1126$ | $0.8366 \pm 0.0478$ | $1.4039 \pm 0.0804$ |
| 16 | **BASE** (OFF) | $0.1766 \pm 0.0279$ | $0.3485 \pm 0.0472$ | $0.1920 \pm 0.0440$ | $0.4066 \pm 0.0525$ |
| | RMIA (OFF) | $0.9645 \pm 0.0050$ | $0.9909 \pm 0.0023$ | $0.9647 \pm 0.0078$ | $0.9918 \pm 0.0024$ |
| | LiRA (OFF) | $-0.0837 \pm 0.0223$ | $-0.0323 \pm 0.0136$ | $-0.0770 \pm 0.0201$ | $-0.0210 \pm 0.0080$ |
| 8 | **BASE** | $0.6116 \pm 0.0601$ | $0.9740 \pm 0.0839$ | $0.9207 \pm 0.0864$ | $1.4146 \pm 0.1155$ |
| | RMIA | $0.9675 \pm 0.0035$ | $0.99322 \pm 0.0015$ | $0.9505 \pm 0.0102$ | $0.9884 \pm 0.0037$ |
| | LiRA | $0.9362 \pm 0.0812$ | $1.5689 \pm 0.1635$ | $1.1176 \pm 0.0898$ | $2.0389 \pm 0.1434$ |
| 4 | **BASE** (OFF) | $0.2312 \pm 0.0364$ | $0.4795 \pm 0.0571$ | $0.24979 \pm 0.0569$ | $0.5437 \pm 0.0737$ |
| | RMIA (OFF) | $0.9698 \pm 0.0032$ | $0.9934 \pm 0.0014$ | $0.9689 \pm 0.0059$ | $0.9942 \pm 0.0021$ |
| | LiRA (OFF) | $-0.0664 \pm 0.0199$ | $-0.0221 \pm 0.0101$ | $-0.0581 \pm 0.0162$ | $-0.0132 \pm 0.0056$ |

Table 5: Attack performance using a threshold estimated by attacking 10 simulated target models. The estimated threshold is the average 1% FPR threshold against the simulated target models. TPR at 1% FPR is reported for comparison. Performance is measured as mean ± standard deviation against 10 target models. The FPR against the target models when using the estimated threshold is close to 1%. Consequently, the TPR at the estimated threshold is close to the TPR at the 1% FPR threshold. This method of estimating the threshold at a given fixed FPR does work for our attacks, LiRA, and RMIA.

| K | ATTACK | CORA (GCN) | | | CITESEER (GAT) | | | PUBMED (GraphSAGE) | | |
| | | TPR@1%FPR (%) | ESTIMATED THRESHOLD | | TPR@1%FPR (%) | ESTIMATED THRESHOLD | | TPR@1%FPR (%) | ESTIMATED THRESHOLD | |
| | | | TPR (%) | FPR (%) | | TPR (%) | FPR (%) | | TPR (%) | FPR (%) |
|---|---|---|---|---|---|---|---|---|---|---|
| 8 | LiRA | $9.25 \pm 3.79$ | $6.97 \pm 0.65$ | $0.83 \pm 0.34$ | $15.35 \pm 3.20$ | $12.78 \pm 0.98$ | $0.78 \pm 0.30$ | $1.22 \pm 0.24$ | $1.24 \pm 0.14$ | $1.04 \pm 0.13$ |
| | RMIA | $15.83 \pm 2.57$ | $16.26 \pm 2.00$ | $1.05 \pm 0.28$ | $24.08 \pm 2.03$ | $21.46 \pm 2.36$ | $0.85 \pm 0.22$ | $3.07 \pm 0.31$ | $2.77 \pm 0.15$ | $0.87 \pm 0.11$ |
| | **BASE** | $\mathbf{15.88 \pm 2.57}$ | $\mathbf{17.19 \pm 1.29}$ | $1.08 \pm 0.39$ | $\mathbf{24.09 \pm 2.03}$ | $\mathbf{22.24 \pm 1.13}$ | $0.87 \pm 0.10$ | $3.07 \pm 0.31$ | $2.87 \pm 0.22$ | $0.90 \pm 0.13$ |
| | **G-BASE** | $14.00 \pm 3.11$ | $14.96 \pm 1.70$ | $1.03 \pm 0.30$ | $21.03 \pm 2.90$ | $21.85 \pm 1.50$ | $1.14 \pm 0.30$ | $\mathbf{5.33 \pm 0.58}$ | $\mathbf{5.39 \pm 0.39}$ | $1.01 \pm 0.12$ |
| 4 | LiRA (OFF) | $12.45 \pm 3.60$ | $11.73 \pm 1.30$ | $0.84 \pm 0.30$ | $18.17 \pm 2.24$ | $20.26 \pm 0.86$ | $1.22 \pm 0.30$ | $1.52 \pm 0.19$ | $2.04 \pm 0.39$ | $1.37 \pm 0.29$ |
| | RMIA (OFF) | $16.94 \pm 3.09$ | $17.39 \pm 1.14$ | $1.09 \pm 0.36$ | $25.38 \pm 3.42$ | $23.87 \pm 0.91$ | $0.89 \pm 0.32$ | $3.41 \pm 0.47$ | $3.14 \pm 0.27$ | $0.88 \pm 0.15$ |
| | **BASE** (OFF) | $\mathbf{17.36 \pm 3.34}$ | $\mathbf{17.58 \pm 1.14}$ | $1.03 \pm 0.30$ | $\mathbf{26.02 \pm 3.38}$ | $\mathbf{24.57 \pm 1.01}$ | $0.97 \pm 0.22$ | $3.30 \pm 0.39$ | $3.17 \pm 0.24$ | $0.91 \pm 0.18$ |
| | **G-BASE** (OFF) | $11.31 \pm 2.97$ | $11.88 \pm 1.18$ | $1.15 \pm 0.45$ | $16.82 \pm 2.72$ | $15.75 \pm 0.80$ | $0.97 \pm 0.31$ | $\mathbf{5.60 \pm 0.40}$ | $\mathbf{5.55 \pm 0.31}$ | $1.00 \pm 0.11$ |

this heuristic rule further, we find that it generally does not hold. In Table 3 and Table 4 we see that the thresholds resulting in an FPR at most 1% or 0.1% are lower than $\beta = 0.99$ or $\beta = 0.999$, respectively, across all datasets. Moreover, Table 6 (graph data) and Table 7 (i.i.d data) show that the actual FPR obtained when setting $\beta = 0.9$ and $\beta = 0.99$ is lower than 10% and 1%, respectively, as would be expected if $\beta = 1 - \alpha$ where to give FPR $\alpha$. Consequently, the TPR obtained at $\beta = 0.9$

Table 6: TPR and FPR at fixed threshold $\beta$ for the RMIA attack with $\gamma = 1$, using the full population as the $Z$ set. Setting $\beta = 1 - \alpha$ does result in a significantly lower FPR than $\alpha$, at the cost of a lower TPR than what is possible to achieve at FPR $\alpha$. 256 shadow models are used for the online attack, and 128 for the offline attack.

| ATTACK | CORA (GCN) | | | | CITESEER (GAT) | | | |
| | $\beta = 0.9$ | | $\beta = 0.99$ | | $\beta = 0.9$ | | $\beta = 0.99$ | |
| | TPR (%) | FPR (%) | TPR (%) | FPR (%) | TPR (%) | FPR (%) | TPR (%) | FPR (%) |
|---|---|---|---|---|---|---|---|---|
| RMIA | $2.73 \pm 0.49$ | $0.00 \pm 0.00$ | $0.00 \pm 0.00$ | $0.00 \pm 0.00$ | $3.31 \pm 0.71$ | $0.01 \pm 0.04$ | $0.00 \pm 0.00$ | $0.00 \pm 0.00$ |
| RMIA (OFF) | $0.00 \pm 0.00$ | $0.00 \pm 0.00$ | $0.00 \pm 0.00$ | $0.00 \pm 0.00$ | $0.10 \pm 0.09$ | $0.00 \pm 0.00$ | $0.00 \pm 0.00$ | $0.00 \pm 0.00$ |

Table 7: TPR and FPR values for RMIA and RMIA (OFF) on CIFAR-10 at decision thresholds $\beta = 1 - \alpha$ where $\alpha \in \{0.01, 0.001\}$. Results are reported as mean $\pm$ standard deviation over 10 runs.

| $K$ | ATTACK | $\beta = 0.99$ | | $\beta = 0.999$ | |
| | | TPR (%) | FPR (%) | TPR (%) | FPR (%) |
|---|---|---|---|---|---|
| 32 | RMIA | $1.90 \pm 0.09$ | $0.11 \pm 0.04$ | $0.20 \pm 0.03$ | $0.00 \pm 0.00$ |
| 16 | RMIA (OFF) | $1.89 \pm 0.11$ | $0.13 \pm 0.06$ | $0.21 \pm 0.03$ | $0.00 \pm 0.01$ |
| 8 | RMIA | $1.82 \pm 0.07$ | $0.18 \pm 0.04$ | $0.20 \pm 0.04$ | $0.01 \pm 0.00$ |
| 4 | RMIA (OFF) | $1.82 \pm 0.09$ | $0.20 \pm 0.05$ | $0.20 \pm 0.02$ | $0.01 \pm 0.01$ |

and $\beta = 0.99$ is significantly lower than what is possible to achieve at FPR 10% and 1%, respectively. As an example, at FPR 1%, online RMIA achieves a mean TPR of 24.00% on Citeseer (see Table 1), whereas using the threshold $\beta = 0.99$ instead results in no true positives at all.

## D  Membership Inference Game for i.i.d. Data

The following definition of a membership inference game closely follows the one in [11, Def. 1] and the ones in [14].

***Definition 5. (Membership inference game)***

1. *The challenger samples a dataset $\mathcal{D}_{train} \subset \mathcal{D}$ from a data population pool $\mathcal{D}$ and trains a model $\boldsymbol{\theta}$ on $\mathcal{D}_{train}$.*

2. *The challenger flips a fair coin to generate a bit b. If $b = 0$, a data point $v$ is randomly selected from $\mathcal{D} \backslash \mathcal{D}_{train}$. If $b = 1$, the data point $v$ is selected from $\mathcal{D}_{train}$.*

3. *The challenger gives the adversary the population pool $\mathcal{D}$, the target sample $v$, and black-box access to the trained model $f_{\boldsymbol{\theta}}$.*

4. *The adversary may also have access to additional side information (such as knowledge about the training algorithm or model architecture). Using this information, the adversary performs a MIA $\hat{m}_v \leftarrow MIA(f_{\boldsymbol{\theta}}, v, \mathcal{D}, \mathcal{H})$, where $\hat{m}_v$ is an estimate of the membership status of sample $v$, $m_v$.*

5. *The attack is successful on a data point $v$ if $\hat{m}_v = m_v$.*

## E  BASE: Practical Bayes-Optimal MIA for i.i.d. Data

### E.1  Proof of Corollary 1

The i.i.d. setting corresponds to $\boldsymbol{A} = \boldsymbol{I}$, where $\boldsymbol{I}$ is the identity matrix. In this case, since the data points are independent, the neighborhood-dependent term (7) vanishes, $\mathcal{G} = \mathcal{D}$, and the membership indicator satisfies $m_v = 1$ if the data sample $v \in [n]$ is included in the training set and $m_v = 0$ otherwise. Hence, $S_{\mathcal{L}}(f_{\boldsymbol{\theta}}, v, \tilde{\mathcal{M}}, \mathcal{D})$ in (6) reduces to the individual loss value of the target sample $v$, i.e., $S_{\mathcal{L}}(f_{\boldsymbol{\theta}}, v, \tilde{\mathcal{M}}, \mathcal{D}) = \ell(f_{\boldsymbol{\theta}}(\boldsymbol{x}_v), \boldsymbol{y}_v)$. Finally, since $p(\boldsymbol{\theta}|m_v, \tilde{\mathcal{M}}, \{x_v\}_{v=1}^n)$ is independent of $m_v$ by assumption, the log-likelihood ratio $\Lambda(\boldsymbol{\theta}, \tilde{\mathcal{M}}, \{x_v\}_{v=1}^n) = \frac{p(\boldsymbol{\theta}|m_v=1, \tilde{\mathcal{M}}, \{x_v\}_{v=1}^n)}{p(\boldsymbol{\theta}|m_v=0, \tilde{\mathcal{M}}, \{x_v\}_{v=1}^n)} = 1$ and it vanishes from the Bayes-optimal membership inference rule (5).

### E.2  BASE: Practical Bayes-Optimal MIA for i.i.d. Data

Recall that under the i.i.d. assumption, we have $\mathcal{G} = \mathcal{D}$ and the loss-based signal simplifies to $S_{\mathcal{L}}(f_{\boldsymbol{\theta}}, v, \tilde{\mathcal{M}}, \mathcal{D}) = \ell(f_{\boldsymbol{\theta}}(\boldsymbol{x}_v), \boldsymbol{y}_v)$ in Theorem 1. Now, since the loss-based signal no longer

depends on $\tilde{\mathcal{M}}$, the only dependence on $\tilde{\mathcal{M}}$ in Bayes-optimal membership inference rule is contained in the posterior model distribution $p(\phi|m_v = 0, \tilde{\mathcal{M}}, \mathcal{D})$. By approximating this posterior model distribution by the prior $p(\phi|\mathcal{D})$, (which we denote simply by $p(\phi)$ to conform to the standard notation), we have removed all the dependence on $\tilde{\mathcal{M}}$ in the approximation and the outer expectation over $\tilde{\mathcal{M}}$ is trivial.

The posterior membership probability $P(m_v = 1|\boldsymbol{\theta}, \mathcal{D})$ then simplifies to

$$P(m_v = 1|\boldsymbol{\theta}, \mathcal{D}) = \sigma\left(-\ell(f_{\boldsymbol{\theta}}(\boldsymbol{x}_v), \boldsymbol{y}_v) - \log \int e^{-\ell(f_\phi(\boldsymbol{x}_v), \boldsymbol{y}_v)} p(\phi)\mathrm{d}\phi + \log\frac{\lambda}{1-\lambda}\right). \quad (17)$$

The remaining expectation over the prior model distribution is still intractable but can be efficiently approximated using Monte Carlo sampling of shadow models,

$$\log \int e^{-\ell(f_{\boldsymbol{\theta}}(\boldsymbol{x}_v), \boldsymbol{y}_v)} p(\phi)\mathrm{d}\phi \approx \log\left(\frac{1}{K}\sum_{k=1}^{K} e^{-\ell(f_{\phi_k}(\boldsymbol{x}_v), \boldsymbol{y}_v)}\right), \quad \phi_k \leftarrow \mathcal{T}(\mathcal{D}_k), \quad (18)$$

where the shadow models are trained on sampled datasets $\mathcal{D}_k$ from the data population, in analogy with (9). Substituting the Monte Carlo approximation from (18) into (17), we can formalize the resulting attack as in Definition 3 (Section 5.2).

Note that neither the sigmoid function nor the membership prior term are necessary for the attack. As shown in Lemma 1, an equivalent attack can be obtained by applying the inverse sigmoid function and subtracting the prior term, which corresponds to a monotonic transformation. However, we retain both the sigmoid function and prior term to preserve the interpretation of the attack score as a posterior probability.

### E.3 Proof of Theorem 2

We begin by establishing the following result.

**Lemma 1.** *Two score-based MIAs are equivalent if their score functions are related by a monotonic transformation.*

*Proof.* Let $\boldsymbol{a}$ and $\boldsymbol{b}$ denote the vector of prediction scores for two MIAs $A$ and $B$, respectively, after attacking an arbitrary target model using $N$ arbitrary target samples. Furthermore, let $\tau_A$ be an arbitrary decision threshold for attack $A$, such that the positive predictions are $\mathcal{M}_A = \{i : i \in \{1, \ldots, N\}, \boldsymbol{a}_i > \tau_A\}$. By assumption, there exists a strictly increasing function $g$ such that $g(\boldsymbol{a}_i) = \boldsymbol{b}_i$ for all $i \in \{1, \ldots, N\}$. Now let $\tau_B = g(\tau_A)$, then the positive predictions of attack B are given by

$$\begin{aligned}
\mathcal{M}_B &= \{i : i \in \{1, N\}, \boldsymbol{b}_i > \tau_B\} \\
&= \{i : i \in \{1, N\}, g(\boldsymbol{a}_i) > g(\tau_A)\} \\
&= \{i : i \in \{1, N\}, \boldsymbol{a}_i > \tau_A\} \\
&= \mathcal{M}_A,
\end{aligned}$$

where the third equality follows from the fact that $g(x) > g(y)$ if and only if $x > y$ for a strictly increasing function $g$. Since $\tau_A$ was arbitrary, $A$ and $B$ are equivalent by Definition 4. $\qquad \square$

The intuition behind the result of Lemma 1 is that a monotonic transformation preserves the order of the membership scores. When a decision threshold is applied to MIA scores, only the target samples with top-$k$ highest score (with $k$ depending on the threshold) are classified as members. However, since the order of the scores is preserved, the top-$k$ scores will correspond to the same target samples for both MIAs.

Equipped with Lemma 1, we are now ready to prove Theorem 2. In particular, since the composition of monotonic transformations is itself a monotonic transformation, it follows that Lemma 1 also applies when the score functions are related by such a composition, which we will use repeatedly in the proof that follows.

The RMIA score function is defined as

$$\Lambda_{\mathrm{RMIA}}(\boldsymbol{x}_i, y_i; \boldsymbol{\theta}) = \Pr_{(\boldsymbol{x}_j, y_j) \sim \pi} \left[ \frac{p(\boldsymbol{\theta}|\boldsymbol{x}_i, y_i)}{p(\boldsymbol{\theta}|\boldsymbol{x}_j, y_j)} \geq \gamma \right], \tag{19}$$

where $\boldsymbol{\theta}$ is the target model parameters, $(\boldsymbol{x}_i, y_i)$ the feature-label pair defining the target sample, and $\pi$ the data population.

To prove that BASE is equivalent to RMIA when $\gamma = 1$, it suffices (by Lemma 1) to show that the their score functions are related by a monotonic transformation. We do this in two steps:

1. We show that BASE is equivalent to an attack that computes the ratio between the target model's confidence value and the expected confidence value over the prior model distribution. We refer to this attack as the *mean confidence attack* (MCA).

2. We derive the monotonic transformation that relates MCA to RMIA, which turns out to be a cumulative distribution function (CDF), restricted to a domain determined by the data population.

The two steps are detailed in the following.

**1.** The BASE score function is given by

$$\Lambda_{\mathrm{BASE}}(\boldsymbol{x}_i, y_i; \boldsymbol{\theta}) = \sigma \left( -\ell(f_{\boldsymbol{\theta}}(\boldsymbol{x}_i), y_i) - \log \int e^{-\ell(f_{\boldsymbol{\phi}}(\boldsymbol{x}_i), y_i)} p(\boldsymbol{\phi}) \mathrm{d}\boldsymbol{\phi} + \log \frac{\lambda}{1 - \lambda} \right). \tag{20}$$

Since the sigmoid function is strictly increasing, it can be removed to obtain an equivalent attack. After removing the sigmoid, the prior term becomes an additive constant, which also defines a monotonic transformation and can therefore be discarded. Applying the (strictly increasing) exponential function to the resulting score function, we obtain

$$\Lambda_{\mathrm{MCA}}(\boldsymbol{x}_i, y_i; \boldsymbol{\theta}) = \frac{f_{\boldsymbol{\theta}}(\boldsymbol{x}_i)_{y_i}}{\int f_{\boldsymbol{\phi}}(\boldsymbol{x}_i)_{y_i} p(\boldsymbol{\phi}) d\boldsymbol{\phi}} = \frac{f_{\boldsymbol{\theta}}(\boldsymbol{x}_i)_{y_i}}{\mathbb{E}_{\boldsymbol{\phi}}[f_{\boldsymbol{\phi}}(\boldsymbol{x}_i)_{y_i}]} \tag{21}$$

where we have used that the confidence is related to the negative log-likelihood loss function by $f_{\boldsymbol{\theta}}(\boldsymbol{x}_i)_{y_i} = e^{-\mathcal{L}(f_{\boldsymbol{\theta}}(\boldsymbol{x}_i), y_i)}$. We note that MCA is a kind of difficulty calibration [37], using the confidence value as the membership score and calibrating by dividing the expected value rather than subtracting it.

**2.** Next, we show that the MCA score in (21) is related to the RMIA score function in (19) when $\gamma = 1$. Applying Bayes' rule to the likelihood ratio in RMIA, we can rewrite the score function as

$$\begin{aligned}
\Lambda_{\mathrm{RMIA}}(\boldsymbol{x}_i, y_i; \boldsymbol{\theta}) &= \Pr_{(\boldsymbol{x}_j, y_j) \sim \pi} \left[ \frac{p(y_i|\boldsymbol{x}_i, \boldsymbol{\theta})p(\boldsymbol{\theta})}{p(y_j|\boldsymbol{x}_j, \boldsymbol{\theta})p(\boldsymbol{\theta})} \frac{p(y_j|\boldsymbol{x}_j)}{p(y_i|\boldsymbol{x}_i)} \geq 1 \right] \\
&= \Pr_{(\boldsymbol{x}_j, y_j) \sim \pi} \left[ \frac{p(y_i|\boldsymbol{x}_i, \boldsymbol{\theta})}{p(y_i|\boldsymbol{x}_i)} \geq \frac{p(y_j|\boldsymbol{x}_j, \boldsymbol{\theta})}{p(y_j|\boldsymbol{x}_j)} \right] \\
&= \Pr_{(\boldsymbol{x}_j, y_j) \sim \pi} \left[ \frac{p(y_i|\boldsymbol{x}_i, \boldsymbol{\theta})}{\mathbb{E}_{\boldsymbol{\phi}}[p(y_i|\boldsymbol{x}_i, \boldsymbol{\phi})]} \geq \frac{p(y_j|\boldsymbol{x}_j, \boldsymbol{\theta})}{\mathbb{E}_{\boldsymbol{\phi}}[p(y_j|\boldsymbol{x}_j, \boldsymbol{\phi})]} \right] \\
&= \Pr_{(\boldsymbol{x}_j, y_j) \sim \pi} \left[ \frac{f_{\boldsymbol{\theta}}(\boldsymbol{x}_i)_{y_i}}{\mathbb{E}_{\boldsymbol{\phi}}[f_{\boldsymbol{\phi}}(\boldsymbol{x}_i)_{y_i}]} \geq \frac{f_{\boldsymbol{\theta}}(\boldsymbol{x}_j)_{y_j}}{\mathbb{E}_{\boldsymbol{\phi}}[f_{\boldsymbol{\phi}}(\boldsymbol{x}_j)_{y_j}]} \right]. \tag{22}
\end{aligned}$$

This expression corresponds to the CDF of the random variable $\boldsymbol{X} = \frac{f_{\boldsymbol{\theta}}(\boldsymbol{x}_j)_{y_j}}{\mathbb{E}_{\boldsymbol{\phi}}[f_{\boldsymbol{\phi}}(\boldsymbol{x}_j)_{y_j}]}$ evaluated at $\Lambda_{\mathrm{MCA}}(\boldsymbol{x}_i, y_i; \boldsymbol{\theta}) = \frac{f_{\boldsymbol{\theta}}(\boldsymbol{x}_i)_{y_i}}{\mathbb{E}_{\boldsymbol{\phi}}[f_{\boldsymbol{\phi}}(\boldsymbol{x}_i)_{y_i}]}$. While a CDF is always non-decreasing, it is not necessarily strictly increasing. However, since the target sample $(\boldsymbol{x}_i, y_i)$ is also part of the data population $\pi$, (22) is strictly increasing on the relevant domain. Specifically, as a function of the real variable $\frac{f_{\boldsymbol{\theta}}(\boldsymbol{x}_i)_{y_i}}{\mathbb{E}_{\boldsymbol{\phi}}[f_{\boldsymbol{\phi}}(\boldsymbol{x}_i)_{y_i}]}$, the CDF (22) can only be constant on intervals where there is no probability mass or density. Therefore, RMIA and MCA are equivalent by Lemma 1.

Combining steps 1 and 2 proves that BASE is equivalent to RMIA with $\gamma = 1$.

## F   Online vs. Offline Attacks

MIAs that use shadow models to estimate distributions of reference models can be designed as online or offline attacks. In the online setting, the adversary can train shadow models on the target sample. This requires no additional assumptions, as the adversary controls shadow model training and can always include the target. However, in certain auditing scenarios, the online setting may be impractical—it would necessitate retraining shadow models for every new audit point. In contrast, the offline setting assumes that shadow models are trained once, without using any target samples. This setting can be of importance when all the target samples are not determined when setting up the attack. However, due to the shadow model training trick introduced in [11] (see also Algorithm 1), it is possible to perform efficient privacy audits in the online setting. Following common terminology, we refer to the shadow models trained on the target sample as *in-models* and otherwise they are referred to as *out-models*.

According to its original paper [13], RMIA is ideally performed in online mode. Consequently, due to the equivalence between BASE and RMIA (see Theorem 2), BASE should also benefit from the use of in-models. However, the Bayes-optimal membership inference formula (5) involves only out-models, consistent with [1]. However, since BASE and G-BASE are only approximations of the Bayes-optimal rule, involving several approximations, we cannot rule out a potential benefit from using in-models in our attack.

Following a similar line of argument as [13], to compensate for the lack of in-models in the offline setting, we introduce a scaling factor $\alpha \in [0, 1]$ on the Monte Carlo loss term over shadow models,

$$P(m_v = 1|\boldsymbol{\theta}, \mathcal{D}) = \sigma\left(-\ell(f_{\boldsymbol{\theta}}(\boldsymbol{x}_v), \boldsymbol{y}_v) - \alpha \log\left(\frac{1}{K}\sum_{k=1}^{K} e^{-\ell(f_{\phi_k}(\boldsymbol{x}_v), \boldsymbol{y}_v)}\right) + \log\frac{\lambda}{1-\lambda}\right). \quad (23)$$

Intuitively, the loss value on the target sample is expected to be smaller when using an in-model compared to an out-model. Thus, the LogSumExp over shadow models is expected to be a negative number of greater magnitude when computed only over out-models in offline mode, as compared to the mix of in- and out-models used in online mode. The scaling factor reduces the magnitude of this term to better match the expected value when using in-models. In principle, this scaling constant could also be used in online mode. We indeed found that this could marginally improve the performance in some cases also in the online setting. However, we believe the small performance benefits do not outweigh the disadvantages of introducing a hyperparamter in need of tuning, and omit from using it in online mode. We also found little to no benefit from introducing it for G-BASE.

## G   Differential Privacy Bound for Bayes-Optimal Membership Inference

It is straightforward to bound the Bayes-optimal membership inference probability in terms of $\epsilon$-differential privacy (DP) [1]. DP is a mathematical framework that defines a notion of privacy for individual data records through a measure of indistinguishability. It was originally proposed in the context of databases, formalizing the intuitive notion that a query function on a database is private if the inclusion or exclusion of a single data record only affects the query output by a small amount.

**Definition 6.** (*$\epsilon$-DP*) *A randomized mechanism $\mathcal{M}$ satisfies $\epsilon$-DP if for any two datasets $D$ and $D'$ differing in a single data sample, and any event $E \subset \mathrm{Range}(\mathcal{M})$, it holds that*

$$\log\frac{P(\mathcal{M}(D) \in E)}{P(\mathcal{M}(D') \in E)} \leq \epsilon. \quad (24)$$

DP is often used as a guarantee for private machine learning. Consider a $\epsilon$-DP training algorithm $\mathcal{T}$ that outputs a set of weights $\boldsymbol{\theta}$ given a training dataset $\mathcal{D}$: $\boldsymbol{\theta} \leftarrow \mathcal{T}(\mathcal{D})$. Given $\mathcal{D}$, the inclusion and exclusion of the target sample $v$ result in two datasets that differ only in one sample. Therefore, the condition for $\epsilon$-DP directly results in the following bound on the Bayes-optimal membership

Table 8: Evaluation of attack performance in the case of a distribution shift in the shadow models. Specifically, the adversary uses a model architecture and training procedure that differs from that of the challenger. Performance is measured in terms of AUC and TPR at 1% and 0.1% FPR. The result is reported as the sample mean $\pm$ the standard deviation over 10 random target models and samples of target nodes. The parameter $K$ denotes the number of shadow models. All attacks undergo a decline in performance compared to the ideal setting where adversary uses the same model architecture and training procedure as the challenger. Our attacks BASE and G-BASE achieves top performance also in this setting.

| K | ATTACK | CORA (GCN) | | | CITESEER (GAT) | | | PUBMED (GRAPHSAGE) | | |
|---|---|---|---|---|---|---|---|---|---|---|
| | | AUC (%) | TPR@FPR (%) | | AUC (%) | TPR@FPR (%) | | AUC (%) | TPR@FPR (%) | |
| | | | 1% | 0.1% | | 1% | 0.1% | | 1% | 0.1% |
| 8 | MLP (0-HOP) | $58.68 \pm 2.33$ | $2.08 \pm 0.99$ | $0.25 \pm 0.30$ | $62.82 \pm 2.37$ | $3.27 \pm 1.47$ | $0.51 \pm 0.55$ | $50.74 \pm 0.49$ | $1.05 \pm 0.18$ | $0.09 \pm 0.07$ |
| | MLP (0+2-HOP) | $60.40 \pm 2.24$ | $2.45 \pm 1.23$ | $0.25 \pm 0.32$ | $64.22 \pm 1.47$ | $2.73 \pm 1.21$ | $0.19 \pm 0.16$ | $50.87 \pm 0.73$ | $1.14 \pm 0.26$ | $0.09 \pm 0.05$ |
| | LIRA | $67.63 \pm 0.88$ | $1.85 \pm 0.97$ | $0.46 \pm 0.58$ | $66.74 \pm 1.41$ | $2.65 \pm 1.35$ | $0.23 \pm 0.31$ | $51.82 \pm 0.57$ | $1.17 \pm 0.23$ | $0.14 \pm 0.10$ |
| | RMIA | $\mathbf{73.86} \pm 1.02$ | $5.97 \pm 1.82$ | $0.81 \pm 0.39$ | $\mathbf{74.96} \pm 1.11$ | $5.34 \pm 1.66$ | $1.14 \pm 0.92$ | $55.02 \pm 0.56$ | $1.50 \pm 0.26$ | $0.18 \pm 0.07$ |
| | BASE | $\mathbf{73.86} \pm 1.02$ | $5.97 \pm 1.82$ | $0.81 \pm 0.39$ | $\mathbf{74.96} \pm 1.11$ | $5.35 \pm 1.66$ | $1.14 \pm 0.92$ | $55.02 \pm 0.56$ | $1.50 \pm 0.26$ | $0.18 \pm 0.07$ |
| | G-BASE (MIA) | $69.34 \pm 1.50$ | $\mathbf{6.71} \pm 2.28$ | $\mathbf{1.43} \pm 1.04$ | $69.83 \pm 0.75$ | $\mathbf{7.48} \pm 1.98$ | $\mathbf{1.30} \pm 1.21$ | $\mathbf{57.09} \pm 0.79$ | $\mathbf{2.38} \pm 0.27$ | $\mathbf{0.42} \pm 0.17$ |
| 4 | LIRA | $71.76 \pm 1.05$ | $2.17 \pm 1.11$ | $0.41 \pm 0.39$ | $73.62 \pm 1.33$ | $3.29 \pm 1.44$ | $0.22 \pm 0.26$ | $52.05 \pm 0.60$ | $1.39 \pm 0.24$ | $0.18 \pm 0.09$ |
| | RMIA | $72.89 \pm 1.08$ | $\mathbf{8.82} \pm 1.70$ | $\mathbf{2.35} \pm 1.46$ | $75.57 \pm 1.07$ | $9.23 \pm 2.99$ | $2.60 \pm 2.12$ | $54.61 \pm 0.64$ | $1.60 \pm 0.28$ | $0.17 \pm 0.11$ |
| | BASE | $\mathbf{73.54} \pm 1.06$ | $8.76 \pm 2.43$ | $2.29 \pm 1.45$ | $\mathbf{76.24} \pm 1.04$ | $\mathbf{10.00} \pm 2.91$ | $\mathbf{3.27} \pm 2.49$ | $54.83 \pm 0.57$ | $1.91 \pm 0.23$ | $0.25 \pm 0.09$ |
| | G-BASE (MIA) | $68.47 \pm 1.66$ | $5.29 \pm 2.01$ | $1.17 \pm 0.65$ | $68.74 \pm 1.22$ | $4.38 \pm 0.77$ | $0.36 \pm 0.51$ | $\mathbf{57.04} \pm 0.65$ | $\mathbf{2.68} \pm 0.29$ | $\mathbf{0.42} \pm 0.09$ |

probability (12):

$$P(m_v = 1 | \boldsymbol{\theta}, \mathcal{D}) = \mathbb{E}_{\tilde{\mathcal{M}} \sim P(\tilde{\mathcal{M}} | \boldsymbol{\theta}, \mathcal{D})} \left[ \sigma \left( \log \frac{p(\boldsymbol{\theta} | m_v = 1, \tilde{\mathcal{M}}, \mathcal{D})}{p(\boldsymbol{\theta} | m_v = 0, \tilde{\mathcal{M}}, \mathcal{D})} + \log \frac{\lambda}{1 - \lambda} \right) \right] \quad (25)$$

$$\leq \mathbb{E}_{\tilde{\mathcal{M}} \sim P(\tilde{\mathcal{M}} | \boldsymbol{\theta}, \mathcal{D})} \left[ \sigma \left( \epsilon + \log \frac{\lambda}{1 - \lambda} \right) \right] \quad (26)$$

$$= \sigma \left( \epsilon + \log \frac{\lambda}{1 - \lambda} \right). \quad (27)$$

As $\epsilon \to 0$, the upper bound approaches $\lambda$. That is, the stronger the DP guarantee, the closer the membership inference to a random guess using only the prior. This $\epsilon$-DP bound is particularly simple when $\lambda = 0.5$, i.e., prior to observing the model, we are maximally uncertain about the membership status of the target sample. In this case, the bound becomes $P(m_v = 1 | \boldsymbol{\theta}, \mathcal{D}) \leq \sigma(\epsilon)$.

## H  Additional Experiments

In this appendix, we present additional experiments and results. Further results for the mismatched adversary setting are presented in Section H.1. We compare our different sampling strategies for G-BASE in Section H.2. Wall-clock time for membership inference across different attacks is reported in Section H.3. In Section H.4, we present further results on the CIFAR-10 and CIFAR-100 datasets, which are widely adopted benchmarks for MIAs on i.i.d. data. Finally, In Section H.5, we include ROC curves in the low FPR regime for some other dataset-target model combinations not presented in Table 1.

### H.1  Robustness to Mismatched Adversary Assumptions

We evaluate the robustness of our attacks and the baseline MIAs in the scenario where the adversary does not have perfect knowledge of the target model architecture and training procedure. In particular, we run the attack using shadow models of a different architecture than the target model, trained using a different optimizer (with hyperparameters tuned for the respective model and optimizer). Specifically, the target models are trained using SGD with momentum, whereas the shadow models are trained using Adam. Moreover, the target model is only trained on 35% of the dataset, whereas the adversary trains the shadow models on 50% of the data by means of the usual shadow model training procedure outlined in Algorithm 1. All models are 2-layer GNNs with the final embedding having as many dimensions as there are classes.

Table 8 shows the attack performance over three different datasets. The shadow model architectures are GAT (with 4 and 2 attention heads in the first and second layer, respectively), GraphSAGE with max aggregation, and GCN, on Cora, Citeseer and Pubmed, respectively.

We observe a decline in attack performance across all attacks compared to the ideal setting (see Table 1). LiRA suffers the most and is not competitive with our attacks BASE and G-BASE.

Table 9: Comparison of different sampling strategies for G-BASE. We also include the performance of G-BASE when the ground-truth sample $\tilde{\mathcal{M}}$ is used (Ground-truth), i.e. the actual target training set mask, indicating how much performance improvements can be made by a more accurate sampling method. Results are reported as mean ± standard deviation over 10 different target models and sets of target nodes. $M = 8$ samples of $\tilde{\mathcal{M}}$ are used in all cases.

| K | SAMPLING METHOD | CORA (GCN) AUC (%) | CORA TPR@FPR (%) 1% | CORA 0.1% | CITESEER (GAT) AUC (%) | CITESEER TPR@FPR (%) 1% | CITESEER 0.1% | PUBMED (GraphSAGE) AUC (%) | PUBMED TPR@FPR (%) 1% | PUBMED 0.1% |
|---|---|---|---|---|---|---|---|---|---|---|
| 8 | Model-independent | 77.15 ± 1.51 | 7.19 ± 3.08 | 0.68 ± 0.92 | 81.78 ± 0.67 | 13.62 ± 2.52 | 1.44 ± 0.89 | 62.92 ± 0.66 | 5.38 ± 0.49 | 1.20 ± 0.35 |
| | 0-hop MIA | 77.42 ± 1.40 | 15.36 ± 3.04 | 5.05 ± 4.60 | 83.31 ± 0.78 | 21.19 ± 4.27 | 6.92 ± 4.61 | 62.96 ± 0.74 | 5.58 ± 1.03 | 1.16 ± 0.49 |
| | Metropolis-Hastings | 76.30 ± 1.30 | 13.23 ± 1.79 | 3.26 ± 2.31 | 83.44 ± 0.83 | 19.82 ± 4.27 | 3.49 ± 2.79 | 63.50 ± 0.71 | 6.10 ± 0.66 | 1.19 ± 0.44 |
| | Gibbs | 76.86 ± 1.61 | 14.02 ± 2.51 | 4.40 ± 2.84 | 84.39 ± 0.67 | 21.38 ± 4.16 | 5.26 ± 3.80 | 64.10 ± 0.69 | 6.28 ± 0.56 | 1.85 ± 0.59 |
| | Ground-truth | 76.52 ± 1.07 | 19.47 ± 3.66 | 7.39 ± 3.92 | 85.27 ± 0.82 | 27.29 ± 3.96 | 8.40 ± 5.35 | 70.63 ± 0.44 | 11.04 ± 0.85 | 4.00 ± 0.69 |
| 4 | Model-independent | 74.25 ± 1.79 | 9.41 ± 2.33 | 3.03 ± 1.67 | 76.26 ± 1.24 | 13.92 ± 2.27 | 4.80 ± 3.24 | 61.97 ± 0.66 | 5.22 ± 0.38 | 1.35 ± 0.30 |
| | 0-hop MIA | 73.60 ± 1.38 | 10.38 ± 2.97 | 3.53 ± 2.06 | 76.65 ± 1.33 | 15.22 ± 2.34 | 4.02 ± 2.36 | 62.06 ± 0.64 | 5.33 ± 0.47 | 1.27 ± 0.26 |
| | Metropolis-Hastings | 73.90 ± 1.24 | 9.97 ± 2.71 | 4.42 ± 1.79 | 77.72 ± 1.00 | 14.68 ± 1.87 | 4.66 ± 2.81 | 62.11 ± 0.80 | 5.30 ± 0.49 | 1.24 ± 0.47 |
| | Gibbs | 73.83 ± 1.74 | 10.21 ± 2.68 | 3.10 ± 1.49 | 77.08 ± 1.19 | 14.69 ± 1.57 | 4.92 ± 3.46 | 62.58 ± 0.64 | 5.59 ± 0.42 | 1.58 ± 0.31 |
| | Ground-truth | 73.00 ± 0.99 | 11.34 ± 2.55 | 2.88 ± 1.49 | 80.06 ± 1.21 | 19.01 ± 2.41 | 6.34 ± 4.10 | 68.09 ± 0.53 | 8.18 ± 0.55 | 2.48 ± 0.39 |

| K | ATTACK | FLICKR (GCN) AUC (%) | FLICKR TPR@FPR (%) 1% | FLICKR 0.1% | AMAZON-PHOTO (GAT) AUC (%) | AMAZON-PHOTO TPR@FPR (%) 1% | AMAZON-PHOTO 0.1% | GITHUB (GraphSAGE) AUC (%) | GITHUB TPR@FPR (%) 1% | GITHUB 0.1% |
|---|---|---|---|---|---|---|---|---|---|---|
| 8 | Model-independent | 60.04 ± 1.01 | 2.82 ± 0.30 | 0.44 ± 0.06 | 56.89 ± 0.45 | 3.70 ± 0.83 | 0.79 ± 0.38 | 57.40 ± 1.09 | 2.93 ± 0.44 | 0.49 ± 0.15 |
| | 0-hop MIA | 57.48 ± 0.55 | 2.51 ± 0.19 | 0.39 ± 0.09 | 56.77 ± 0.85 | 3.99 ± 0.62 | 0.82 ± 0.47 | 57.38 ± 1.40 | 3.00 ± 0.52 | 0.55 ± 0.18 |
| | Metropolis-Hastings | 56.98 ± 0.71 | 2.17 ± 0.24 | 0.29 ± 0.08 | 56.31 ± 0.86 | 3.37 ± 0.58 | 0.62 ± 0.40 | 56.35 ± 1.11 | 2.58 ± 0.55 | 0.39 ± 0.21 |
| | Gibbs | 57.88 ± 0.91 | 2.33 ± 0.27 | 0.37 ± 0.15 | 56.53 ± 0.64 | 4.44 ± 0.60 | 0.76 ± 0.49 | 57.47 ± 1.43 | 3.18 ± 0.54 | 0.55 ± 0.17 |
| | Ground-truth | 69.95 ± 1.79 | 8.58 ± 1.39 | 2.15 ± 0.50 | 61.98 ± 1.34 | 8.37 ± 1.34 | 2.99 ± 0.48 | 71.64 ± 3.99 | 11.58 ± 3.34 | 4.56 ± 2.03 |
| 4 | Model-independent | 60.04 ± 1.01 | 2.88 ± 0.26 | 0.50 ± 0.06 | 57.81 ± 0.71 | 4.94 ± 0.78 | 1.63 ± 0.44 | 57.78 ± 1.22 | 3.10 ± 0.56 | 0.58 ± 0.15 |
| | 0-hop MIA | 57.71 ± 0.66 | 2.48 ± 0.21 | 0.40 ± 0.07 | 57.74 ± 0.85 | 4.82 ± 0.88 | 1.38 ± 0.37 | 57.72 ± 1.06 | 3.12 ± 0.47 | 0.60 ± 0.19 |
| | Metropolis-Hastings | 56.99 ± 0.75 | 1.99 ± 0.22 | 0.29 ± 0.09 | 56.25 ± 1.11 | 3.76 ± 0.68 | 1.10 ± 0.59 | 56.37 ± 1.12 | 2.62 ± 0.64 | 0.47 ± 0.28 |
| | Gibbs | 58.63 ± 0.95 | 2.48 ± 0.32 | 0.37 ± 0.07 | 57.94 ± 0.54 | 4.61 ± 1.00 | 1.69 ± 0.82 | 58.16 ± 1.48 | 3.44 ± 0.64 | 0.73 ± 0.26 |
| | Ground-truth | 69.70 ± 1.83 | 7.54 ± 1.19 | 1.46 ± 0.31 | 62.43 ± 1.33 | 8.06 ± 1.34 | 2.81 ± 1.28 | 70.46 ± 3.78 | 9.61 ± 2.85 | 3.38 ± 1.71 |

## H.2 Comparison of Sampling Strategies for G-BASE

In Section 4.3 and Appendix B.2, we proposed four different sampling strategies to sample from $P(\tilde{\mathcal{M}}|\boldsymbol{\theta}, \mathcal{G})$, for the purpose of evaluating the outer Monte Carlo estimate in the G-BASE attack Definition 2. Here, we further evaluate and compare the impact of the sampling strategy on the attack performance of G-BASE. Table 9 shows the attack performance of G-BASE over different datasets and model architectures, using each of our four sampling strategies; model-independent sampling, 0-hop MIA sampling (using BASE), Metropolis-Hastings sampling, and Gibbs sampling. The Metropolis-Hastings step size parameter $\epsilon$, controlling the fraction of indicator variables that are flipped in each step, is chosen such that the acceptance rate is in the range $0.2$-$0.4$. We run the Gibbs sampling for one iteration through all the indicator variables. Note that for 0-hop MIA sampling, the same set of shadow models used for BASE (to obtain membership probabilities for this sampling strategy), can be used for G-BASE, and the membership probabilities can be computed once, before running the attack.

To gain insight into how much more the attack is able to gain from a more accurate sampling method, we also run G-BASE using the ground-truth target training set as sample $\tilde{\mathcal{M}}$. Surprisingly, using the ground-truth $\tilde{\mathcal{M}}$ does not help the G-BASE attack against a GCN target model on Cora. However, on other datasets, the ground-truth sample improves the performance significantly, indicating that a better sampling strategy can still make the attack more effective. We also note that, in many cases, the choice of sampling strategy has only a modest effect on overall attack performance, particularly for low values of $M$. The precision of the sampled graphs relative to the ground-truth target training graph generally improves only slightly when using model-dependent sampling, which explains the similar performance observed between model-independent sampling and more sophisticated methods. However, there are instances where the sampling method has a more pronounced impact—for example, model-independent sampling on Flickr.

Recall that we have effectively approximated the model distribution $p(\phi|m_v = 0, \tilde{\mathcal{M}}, \mathcal{G})$ by a distribution $p(\phi|\mathcal{G})$, independent of $v$ and $\tilde{\mathcal{M}}$, when reusing the same set of shadow models for each target node. A consequence of this approximation is that even when obtaining highly fidelity samples $\tilde{\mathcal{M}} \sim P(\tilde{\mathcal{M}}|\boldsymbol{\theta}, \mathcal{G})$, only the loss signal for the target model is precise, and the loss signals for the shadow models will contain a lot of noise due to the mixture of member and non-member nodes in their local neighborhood around the target node. If $p(\phi|m_v = 0, \tilde{\mathcal{M}}, \mathcal{G})$ were estimated more accurately, e.g., by training shadow models on the nodes masked by $\tilde{\mathcal{M}}$, then the fidelity of the sampling strategy is expected to have a larger impact on the attack performance.

Table 10: Performance of G-BASE under varying $M$ (number of samples $\tilde{\mathcal{M}}$). The sampling strategy is model-independent, 0-hop MIA sampling, and Gibbs sampling, for Flickr, Amazon-Photo, Pubmed, respectively. The attack performance consistently increases with an increasing $M$, but with signs of diminishing returns.

| MODE | $K$ | $M$ | FLICKR (GCN) | | | AMAZON-PHOTO (GAT) | | | PUBMED (GRAPHSAGE) | | |
|---|---|---|---|---|---|---|---|---|---|---|---|
| | | | AUC (%) | TPR@FPR (%) | | AUC (%) | TPR@FPR (%) | | AUC (%) | TPR@FPR (%) | |
| | | | | 1% | 0.1% | | 1% | 0.1% | | 1% | 0.1% |
| ONLINE | 8 | 4 | $59.12 \pm 1.00$ | $2.57 \pm 0.34$ | $0.41 \pm 0.06$ | $56.38 \pm 0.66$ | $3.27 \pm 0.73$ | $0.49 \pm 0.39$ | $63.76 \pm 0.51$ | $5.99 \pm 0.65$ | $1.64 \pm 0.47$ |
| | | 8 | $60.04 \pm 1.02$ | $2.89 \pm 0.33$ | $0.45 \pm 0.11$ | $57.02 \pm 0.60$ | $3.74 \pm 0.84$ | $0.69 \pm 0.25$ | $64.08 \pm 0.67$ | $6.16 \pm 0.62$ | $1.80 \pm 0.67$ |
| | | 16 | $60.60 \pm 1.13$ | $3.08 \pm 0.34$ | $0.49 \pm 0.11$ | $57.00 \pm 0.89$ | $4.24 \pm 1.09$ | $0.84 \pm 0.40$ | $64.54 \pm 0.58$ | $6.68 \pm 0.66$ | $1.87 \pm 0.65$ |
| | | 32 | $60.97 \pm 1.17$ | $3.19 \pm 0.40$ | $0.52 \pm 0.13$ | $57.23 \pm 0.94$ | $4.26 \pm 0.82$ | $1.26 \pm 0.70$ | $64.74 \pm 0.47$ | $6.71 \pm 0.70$ | $2.13 \pm 0.64$ |
| OFFLINE | 4 | 4 | $59.07 \pm 0.88$ | $2.65 \pm 0.27$ | $0.43 \pm 0.08$ | $57.00 \pm 0.89$ | $4.16 \pm 0.94$ | $1.33 \pm 0.44$ | $62.18 \pm 0.62$ | $5.49 \pm 0.55$ | $1.40 \pm 0.28$ |
| | | 8 | $60.09 \pm 0.95$ | $2.92 \pm 0.38$ | $0.45 \pm 0.13$ | $57.94 \pm 0.78$ | $4.78 \pm 1.25$ | $1.39 \pm 0.57$ | $62.87 \pm 0.53$ | $5.78 \pm 0.67$ | $1.51 \pm 0.52$ |
| | | 16 | $60.60 \pm 1.13$ | $3.11 \pm 0.24$ | $0.52 \pm 0.10$ | $57.99 \pm 1.11$ | $4.68 \pm 0.95$ | $1.53 \pm 0.58$ | $62.84 \pm 0.65$ | $5.80 \pm 0.38$ | $1.45 \pm 0.42$ |
| | | 32 | $61.02 \pm 1.14$ | $3.16 \pm 0.42$ | $0.54 \pm 0.10$ | $58.53 \pm 1.08$ | $4.98 \pm 1.13$ | $1.68 \pm 0.56$ | $63.02 \pm 0.64$ | $5.94 \pm 0.42$ | $1.50 \pm 0.27$ |

Table 11: Comparison of wall-clock time for the inference phase of our attacks, RMIA, and LiRA.

| $K$ | ATTACK | WALL-CLOCK TIME (S) | | |
|---|---|---|---|---|
| | | FLICKR (GCN) | AMAZON-PHOTO (GAT) | PUBMED (GRAPHSAGE) |
| 8 | LiRA | $0.02585 \pm 0.02018$ | $0.02041 \pm 0.01970$ | $0.01382 \pm 0.01850$ |
| | RMIA | $2.04167 \pm 0.01045$ | $0.19090 \pm 0.00074$ | $0.50807 \pm 0.00149$ |
| | **BASE** | $0.01488 \pm 0.00004$ | $0.01096 \pm 0.00008$ | $0.00444 \pm 0.00003$ |
| | **G-BASE** (MIA) | $6182.23 \pm 342.81$ | $680.38 \pm 30.55$ | $684.11 \pm 146.37$ |
| | **G-BASE** (GIBBS) | $18377.06 \pm 484.46$ | $2013.80 \pm 11.32$ | $1944.71 \pm 318.62$ |
| 4 | LiRA | $0.01964 \pm 0.00026$ | $0.01361 \pm 0.00030$ | $0.00730 \pm 0.00028$ |
| | RMIA | $2.04350 \pm 0.01280$ | $0.19179 \pm 0.00101$ | $0.50834 \pm 0.00102$ |
| | **BASE** | $0.01508 \pm 0.00004$ | $0.01119 \pm 0.00007$ | $0.00468 \pm 0.00004$ |
| | **G-BASE** (MIA) | $4203.32 \pm 198.85$ | $416.16 \pm 19.29$ | $408.95 \pm 6.72$ |
| | **G-BASE** (GIBBS) | $12055.83 \pm 576.11$ | $1262.03 \pm 20.53$ | $1233.48 \pm 17.46$ |

**How many graph samples is enough?** The sampled graphs $\tilde{\mathcal{M}}$ are used to compute a Monte Carlo estimation of the outer expectation of the Bayes-optimal membership inference rule (5). We therefore expect the attack performance to improve with an increasing $M$. In practice, however, we are constrained by the available computational resources. To get insight into how many samples are required to get a satisfactory performance, we run G-BASE with varying $M$ over various target models and different sampling strategies. Table 10 reports the measured attack performance. In particular, we use G-BASE with model-independent sampling, 0-hop MIA sampling, and Gibbs sampling for the Flicker, Amazon-Photo, and Pubmed target model, respectively. Although we consistently observe an increasing attack performance as $M$ increases, there are signs of diminishing returns.

## H.3 Wall-Clock Time of Membership Inference

To quantify the computational efficiency of our attacks, LiRA and RMIA, we measure wall-clock time for their inference phase. Table 11 reports these times for various target models across multiple datasets and GNN architectures. For G-BASE, we measure inference time under both 0-hop MIA sampling (using BASE as the 0-hop MIA) and Gibbs sampling.

Among all attacks, BASE is the most efficient—approximately 20 to 200 times faster than RMIA while achieving comparable performance. The results also show that G-BASE is orders of magnitude slower, primarily due to the lack of parallelization across target nodes. Nevertheless, its inference time remains tractable. We believe it is important to accurately evaluate the data leakage associated with membership inference, even if this comes with a higher computational cost.

## H.4 i.i.d. Data

To demonstrate BASE on i.i.d. data, we train a Wide ResNet [38] with depth 28 and width 2 on both CIFAR-10 and CIFAR-100 for 100 epochs using standard data augmentations and early stopping. Each experiment is averaged across ten target models. For CIFAR-10, the target model achieves a mean training accuracy of $93.73\% \pm 1.15\%$ and a test accuracy of $80.45\% \pm 1.53\%$. For CIFAR-100, the model reaches a training accuracy of $75.97\% \pm 6.68\%$ and a test accuracy of $49.89\% \pm 1.88\%$.

Table 12: Comparison of different attacks on Wide ResNet-28-2 trained on CIFAR-10 and CIFAR-100. Performance is measured in terms of AUC and TPR at 1% and 0.1% FPR, and the results are reported as mean ± standard deviation over 10 random target models.

| | | CIFAR-10 | | | CIFAR-100 | | |
|---|---|---|---|---|---|---|---|
| | | | TPR@FPR (%) | | | TPR@FPR (%) | |
| K | ATTACK | AUC | 1% | 0.1% | AUC | 1% | 0.1% |
| 32 | **BASE** | $\mathbf{62.94 \pm 2.06}$ | $\mathbf{5.92 \pm 1.27}$ | $\mathbf{1.66 \pm 0.43}$ | $\mathbf{74.80 \pm 3.22}$ | $\mathbf{12.08 \pm 3.35}$ | $\mathbf{4.11 \pm 1.69}$ |
| | **RMIA** | $\mathbf{62.94 \pm 2.06}$ | $\mathbf{5.92 \pm 1.27}$ | $\mathbf{1.66 \pm 0.43}$ | $\mathbf{74.80 \pm 3.22}$ | $\mathbf{12.08 \pm 3.35}$ | $\mathbf{4.11 \pm 1.69}$ |
| | LIRA | $61.00 \pm 1.45$ | $5.21 \pm 0.91$ | $1.32 \pm 0.33$ | $72.65 \pm 1.80$ | $9.99 \pm 1.86$ | $2.47 \pm 0.87$ |
| 16 | BASE (OFF) | $61.67 \pm 2.10$ | $\mathbf{5.67 \pm 1.29}$ | $\mathbf{1.64 \pm 0.39}$ | $71.35 \pm 3.36$ | $\mathbf{8.89 \pm 2.97}$ | $\mathbf{3.07 \pm 1.42}$ |
| | RMIA (OFF) | $\mathbf{62.45 \pm 2.10}$ | $5.61 \pm 1.23$ | $1.57 \pm 0.40$ | $70.65 \pm 3.30$ | $7.35 \pm 2.21$ | $2.07 \pm 0.94$ |
| | LIRA (OFF) | $60.51 \pm 1.75$ | $4.10 \pm 0.61$ | $0.94 \pm 0.25$ | $\mathbf{72.33 \pm 2.66}$ | $8.36 \pm 1.65$ | $1.60 \pm 0.69$ |
| 8 | **BASE** | $\mathbf{62.64 \pm 1.96}$ | $\mathbf{5.21 \pm 0.99}$ | $\mathbf{1.19 \pm 0.38}$ | $\mathbf{73.98 \pm 3.18}$ | $\mathbf{10.34 \pm 2.91}$ | $\mathbf{2.81 \pm 0.95}$ |
| | **RMIA** | $\mathbf{62.64 \pm 1.96}$ | $\mathbf{5.21 \pm 0.99}$ | $\mathbf{1.19 \pm 0.38}$ | $\mathbf{73.98 \pm 3.18}$ | $\mathbf{10.34 \pm 2.91}$ | $\mathbf{2.81 \pm 0.95}$ |
| | LIRA | $58.79 \pm 0.99$ | $3.64 \pm 0.67$ | $0.85 \pm 0.25$ | $69.22 \pm 1.60$ | $7.29 \pm 1.37$ | $1.60 \pm 0.58$ |
| 4 | BASE (OFF) | $61.54 \pm 1.98$ | $\mathbf{5.05 \pm 1.06}$ | $\mathbf{1.23 \pm 0.39}$ | $71.26 \pm 3.28$ | $\mathbf{8.23 \pm 2.62}$ | $\mathbf{2.45 \pm 1.20}$ |
| | RMIA (OFF) | $\mathbf{62.17 \pm 1.96}$ | $4.94 \pm 1.07$ | $1.19 \pm 0.39$ | $70.48 \pm 3.25$ | $6.45 \pm 1.88$ | $1.45 \pm 0.67$ |
| | LIRA (OFF) | $59.80 \pm 1.67$ | $3.77 \pm 0.76$ | $0.78 \pm 0.24$ | $71.00 \pm 2.68$ | $7.27 \pm 1.45$ | $1.26 \pm 0.57$ |

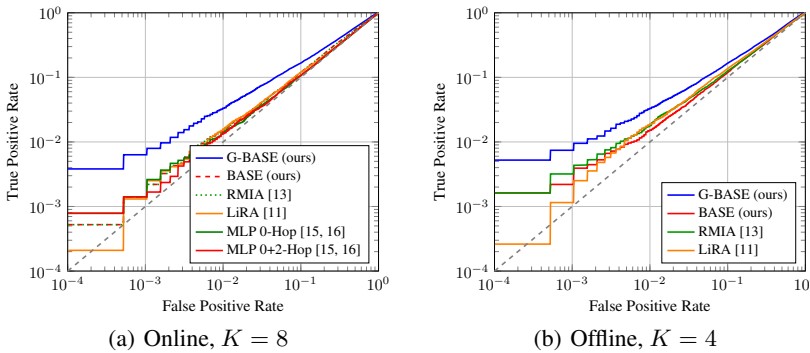

(a) Online, $K = 8$      (b) Offline, $K = 4$

Figure 4: Average ROC curves (10 runs) for the Amazon-Photo dataset with GraphSAGE as target model.

Table 12 presents the attack performance of BASE, RMIA, and LiRA. In the online setting, BASE and RMIA exhibit identical performance, in line with Theorem 2. Both methods consistently outperform LiRA across the evaluated configurations. In the offline setting, while the differences are less pronounced, BASE achieves superior performance over both RMIA and LiRA at low false positive rates.

## H.5 ROC curves

To compare the attack performances in terms of TPR over all low FPRs, we provide average ROC curves in Figures 4 to 7. $K$ denotes the number of shadow models used. We see that in the online setting, the ROC curves of BASE and RMIA are identical, as is expected in light of Theorem 2 (we use $\gamma = 1$ in RMIA). In the offline setting, BASE and RMIA perform similarly, but not equivalently. G-BASE achieves the best performance in terms of TPR at low FPR in all cases.

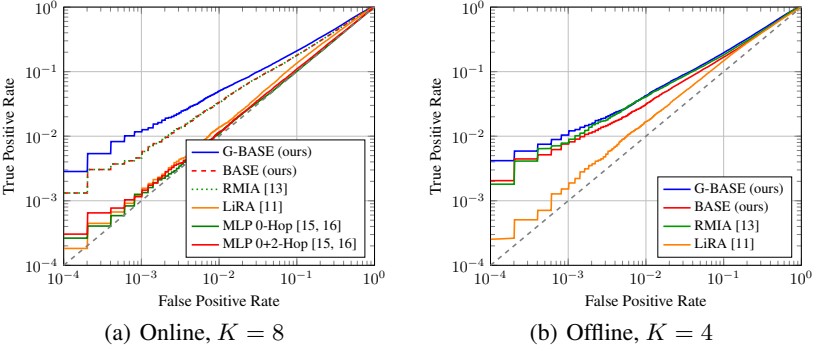

(a) Online, $K = 8$      (b) Offline, $K = 4$

Figure 5: Average ROC curves (10 runs) for the PubMed dataset with GCN as target model.

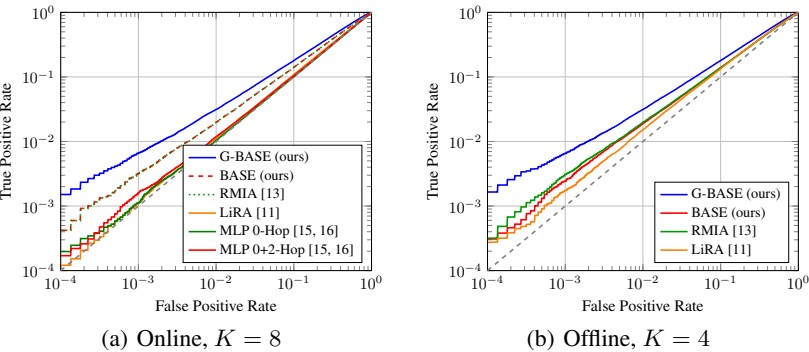

(a) Online, $K = 8$      (b) Offline, $K = 4$

Figure 6: Average ROC curves (10 runs) for the Flickr dataset with GAT as target model.

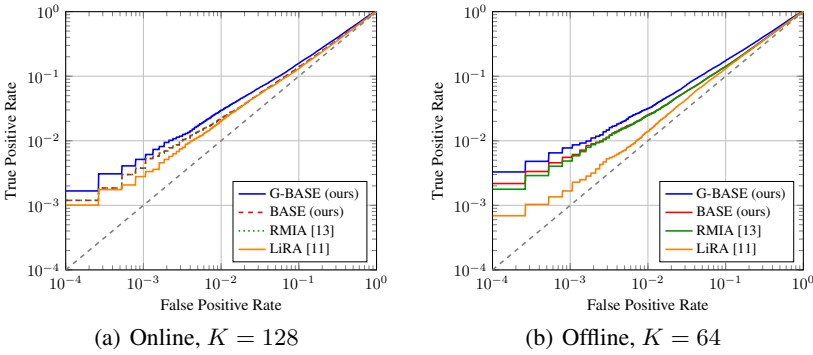

(a) Online, $K = 128$      (b) Offline, $K = 64$

Figure 7: Average ROC curves (10 runs) for the Github dataset with GraphSAGE as target model.

