# OpenReview forum: "Practical Bayes-Optimal Membership Inference Attacks"
_NeurIPS.cc/2025/Conference — NeurIPS 2025 poster_

### Official Review · Reviewer_U2Mu · 2025-06-23

**Clarity:** 3
**Significance:** 3
**Originality:** 4
**Rating:** 5
**Confidence:** 5

**Summary:**

This paper proposes a theoretically grounded and practical approach to membership inference attacks (MIAs) for both i.i.d. and graph-structured data. Building on a Bayesian decision-theoretic framework, the authors derive a Bayes-optimal inference rule for node-level MIAs on graph neural networks (GNNs), addressing a previously unresolved question in privacy auditing. To render the intractable Bayes-optimal rule usable in practice, the paper introduces BASE (for i.i.d. data) and G-BASE (for graph data) as efficient approximations. These methods outperform or match existing state-of-the-art approaches like LIRA and RMIA, while requiring significantly fewer computational resources. The authors also reveal that RMIA is a special case of BASE under specific conditions, thus offering a principled interpretation of RMIA.

**Questions:**

1. In Section 4.1, the adversary is assumed to know the full graph G. How realistic is this in practical scenarios, especially for large or proprietary graphs? Could the method be adapted to a setting with partial or noisy knowledge of G?
2. While the paper claims BASE and G-BASE are more efficient than RMIA/LIRA, a direct analysis of computational cost (e.g., wall-clock time or FLOPs) is missing. Including such data would strengthen the practical contribution.
3. The performance of G-BASE under different shadow model counts or sampling methods is briefly mentioned. Could the authors provide more systematic analysis or guidance on selecting these parameters?
4. The work focuses on node-level MIAs for GNNs. Could the Bayesian framework be extended to edge- or graph-level MIAs? If so, how might the challenges differ?
5. Given the probabilistic nature of the Bayes-optimal rule, have the authors explored confidence calibration of their predictions? This could have practical value in privacy auditing tools.

**Ethical Concerns:**

["NO or VERY MINOR ethics concerns only"]

**Final Justification:**

Based on previous comments and the author's response, I keep my score.

**Limitations:**

Partially addressed. While the authors acknowledge the intractability of the Bayes-optimal rule and propose efficient approximations, the discussion of potential negative societal impact (e.g., how such attacks may be misused) is limited. I encourage the authors to elaborate on possible misuse scenarios and mitigation strategies, especially in light of GDPR concerns.

**Paper Formatting Concerns:**

None.

**Quality:**

4

**Strengths And Weaknesses:**

Strengths:
1. The paper derives a novel Bayes-optimal MIA for graph-structured data, filling a theoretical gap in the literature. The generalization of from i.i.d. to graph settings is significant and well-justified.
2. The proposed approximations BASE and G-BASE are computationally efficient and outperform existing methods in many settings, making them attractive for real-world privacy auditing.
3. The paper is well-organized with clear definitions, formulations, and diagrams. The step-by-step derivation of the Bayes-optimal rule and approximations is particularly helpful.

Weaknesses:
1. The assumption that the adversary has access to the full graph G may limit the applicability in real-world scenarios, especially for GNN attacks. This assumption should be more thoroughly discussed or relaxed.
2. The empirical performance of G-BASE may depend on hyperparameters such as the number of shadow models and sampling strategies. More discussion on robustness to such factors would be helpful.
3. While the work has implications for data privacy, the broader ethical and societal implications (e.g., how adversaries might exploit such tools) are not sufficiently addressed.

---

> ### Author Rebuttal · Authors · 2025-07-31
>
> We thank the reviewer for the thoughtful and detailed feedback, which has helped us improve the paper. We hope we have addressed all of your comments thoroughly and satisfactorily.
>
> ## WEAKNESSES
> ### W1.
> See response to your Question 1 below.
>
> ### W2.
> See response to your Question 3 below.
>
> ### W3.
> We appreciate the reviewer’s comment and agree that the ethical and societal implications of privacy attacks warrant explicit discussion. Our work focuses on quantifying privacy risks in machine learning models, and we recognize that the same techniques could, in principle, be used maliciously. However, we align with the long-standing practice in the cryptography and security communities, where publishing attacks is essential to understanding system vulnerabilities and ultimately designing safer, more trustworthy systems.
>
> Our stance is that the benefit of making such vulnerabilities visible (specifically enabling developers and auditors to assess membership leakage before releasing models trained on sensitive data) far outweighs the potential misuse of these attacks. We believe that transparency is a prerequisite for responsible deployment, and our contribution is intended to support risk-aware decision-making and the development of more robust privacy-preserving techniques.
>
> We will add a brief discussion to the camera-ready version to reflect this perspective more clearly.
>
> ## QUESTIONS
> ### Q1.
> We agree that assuming the adversary has access to the full graph
> $\mathcal{G}$ is a strong assumption. However, this assumption is conceptually similar to the standard membership inference threat model (e.g., in the LIRA [1] and RMIA [2] papers), where the adversary is assumed to have access to a population pool from which the training set is sampled.  Similarly, in our setting, full graph access corresponds to a worst-case assumption, which is particularly relevant for privacy auditing.
>
> In our setting, when the challenger samples a relatively small subset of nodes from a large population graph, this threat model remains relevant for practical scenarios. In our experiments (except in the mismatched adversary scenario), we assume the adversary has perfect side knowledge about the fraction of nodes sampled by the challenger (which mirrors common assumptions in the i.i.d. MIA literature) and let the challenger sample 50\% of the nodes. Compared to the case where a smaller fraction is sampled, this setup is favorable to the adversary, as it increases the likelihood of overlap between the shadow and target training sets. Accordingly, our reported attack performance should be interpreted as an upper bound. Such upper bounds are valuable in privacy auditing, as they help assess worst-case exposure under strong adversarial assumptions.
>
> We agree that relaxing the assumption of full graph knowledge, e.g., allowing the adversary to observe only a noisy or partially observed graph, is an interesting direction for future work. For clarity and simplicity, and privacy auditing purposes, we focused on the full-graph setting in this work. That said, we see extensions to weaker threat models as both feasible and worthwhile, but they are non-trivial and would require substantial work and careful thought.
>
> [1] Carlini et al. "Membership inference attacks from first principles." IEEE SP, 2022.
>
> [2] Zarifzadeh et al. "Low-cost high-power membership inference attacks", ICML 2024.
>
> ### Q2.
> Thank you for your comment. We acknowledge that a more detailed analysis of the computational complexity of the proposed attacks would strengthen the paper and plan to include this discussion in the revised manuscript.
>
> To clarify, our claim on computational efficiency refers partly to BASE and G-BASE being feasible approximations of the intractable Bayes-optimal membership inference rule, and partly to BASE being a more efficient option than RMIA and LiRA. We realize that the current phrasing of some statements including both BASE and G-BASE may have unintentionally implied otherwise, and we will revise it to avoid any misunderstanding.
>
> BASE is more efficient than RMIA, which is the current state-of-the-art method, because it requires significantly fewer queries to the target model during the attack. More specifically,
> our attack BASE requires only a single query to the target model, while RMIA may require thousands. LiRA has similar per-shadow-model efficiency as BASE (in terms of model queries), but our results show that BASE achieves strong performance with relatively few shadow models, while LiRA typically requires a larger number, as also noted in the RMIA paper [2].
>
> On the other hand, G-BASE is more computationally demanding. This is expected, as it is specifically designed to leverage the full graph structure to approximate the Bayes-optimal membership inference rule for GNNs (derived in Sec. 4.2), which naturally introduces additional cost. This sets it apart from BASE, RMIA, and LiRA, which cannot exploit graph connectivity and thus do not capture this richer structure. Unlike BASE, which can evaluate all target nodes in one pass, G-BASE must process each node individually, since its loss signal requires sampling subgraphs and comparing losses with and without the target node for each sample.
>
> Despite its higher cost, G-BASE remains a feasible and effective option for privacy auditing of GNNs, significantly outperforming baseline methods, including BASE.
>
> We agree that a direct analysis of computational cost would strengthen this claim, and we have measured the average wall-clock time (in seconds) for BASE and RMIA on the Pubmed and Flickr datasets:
>
> | | Pubmed | Flickr |
> |----------|:----------:|:----------:|
> | **BASE**  | 0.0065  | 0.0898 |
> | **RMIA** | 0.4992  | 2.1735  |
> | **LiRA** | 2.3618 | 11.1613 |
>
> The time measured here is the inference time of the (online) attacks, after 8 shadow models has been trained. The time to train the shadow models is the same for all attacks. We will incorporate these results into the camera-ready version to support our efficiency claim more concretely.
>
> ### Q3.
> We agree that providing more guidance on the selection of shadow model count and sampling strategy would strengthen the paper, and we will clarify this in the revised version.
>
> In general, increasing the number of shadow models improves attack performance by providing a better approximation of the model distribution. However, this comes with additional computational cost. In practice, we recommend using as many shadow models as computational resources allow.
> However, our experiments show that BASE and G-BASE achieve strong performance even with a relatively small number of shadow models, with performance improving only marginally when increasing the number of shadow models from 8 to 128. This is in contrast to LiRA [1], which typically requires significantly more shadow models, as already noted in the RMIA paper [2].
>
> Regarding sampling strategies, Appendix H.3 presents a detailed comparison and shows that G-BASE performs consistently well across all three proposed sampling methods. We attribute this robustness to the way G-BASE approximates the Bayes-optimal decision rule. For the reviewers convenience, we provide an excerpt from Appendix H.3:
>
> "Recall that we have approximated the model distribution $p(\phi|\tilde{\mathcal{M}},\mathcal{G})$ by a prior distribution $p(\phi|\mathcal{G})$. If  $p(\phi|\tilde{\mathcal{M}},\mathcal{G})$ were estimated more accurately, e.g., by training shadow models on the nodes indicated by $\tilde{\mathcal{M}}$, then the fidelity of the sampling strategy may have a larger impact on the attack performance."
>
> We will clarify these points in the revised version of the paper to provide more practical guidance.
>
> ### Q4.
> We agree that extending the Bayesian framework to other GNN tasks, such as graph-level and edge-level MIAs, is a natural and promising direction for future work.
>
> Extending the Bayesian framework to graph-level MIAs is relatively straighforward. Standard graph classification datasets typically assume i.i.d. samples at the graph level, making them compatible with the assumptions of the Bayes-optimal framework for i.i.d. data. As a result, this setting can be addressed similarly to traditional MIAs on i.i.d. data, see, e.g., [3].
>
> In contrast, extending the framework to edge-level MIAs poses more significant  challenges. Formulating a proper threat model and identifying informative attack signals at the edge level is nontrivial. We are currently investigating edge-level MIAs as part of our ongoing research and consider this a promising and largely unexplored area.
>
> [3] Krüger et al. "Publishing neural networks in drug discovery might compromise training data privacy." Journal of Cheminformatics, 2025.
>
> ### Q5.
> We thank the reviewer for this thoughtful and constructive suggestion. Indeed, given the probabilistic nature of our scoring function, applying confidence calibration to the output of the attack is a natural and valuable extension. While our Bayes-optimal framework produces scores that can be interpreted as membership probabilities, these may be miscalibrated in practice. Calibrating these outputs using methods such as Platt scaling or isotonic regression [4] could make the scores more reflective of true membership likelihoods. This would significantly enhance the interpretability and practical usefulness of the attack in privacy auditing scenarios, where stakeholders may rely on well-calibrated risk scores to assess or mitigate exposure. We had not explored this direction prior to the review, and we appreciate the insight. We will mention this point in the final version as a promising direction for future work.
>
> [4] Guo et al. "On calibration of modern neural networks." ICML, 2017.

---

### Official Review · Reviewer_CHe9 · 2025-07-03

**Clarity:** 2
**Significance:** 3
**Originality:** 2
**Rating:** 4
**Confidence:** 3

**Summary:**

This paper introduces BASE and G-BASE, two membership inference attacks (MIAs) for iid and node-level graph data, where an adversary aims to identify whether a specific data point is included in the training set of a given trained model. These methods are developed by approximating the Bayes optimal membership inference rule. The evaluation on common graph benchmarks shows that these methods work better or on par with the statistics-based methods like LIRA or RMIA, most importantly, the authors claim that the developed methods have a lower computational cost.

**Questions:**

Major concerns/questions:
1. In the approximation of the integral (equation 8), $S_L$ function requires $\tilde{M}$ and not $m_v$. It is unclear how $m_v$ is used here. Please clarify the substitution.
2. In definition 2 of G-BASE attack, the dependence on index i in $\tilde{M}$ and in $S_L$ appears to be missing. The current definition, although implemented and it seemed to work experimentally, clarification is needed on whether the sampling is correctly reflected in the current form.
3. The posterior is approximated with its prior in line 218. How to pick an appropriate prior $p(\phi \mid \mathcal{G})$? The choice is not trivial, and a discussion on it with practical justification is needed.
4. It is not clearly explained how LIRA and RMIA (methods for iid data) are evaluated for graphs in the paper.
5. Line 308: it is not clear why a different number of sampled graphs is chosen and not equal to the number of shadow models?
6. Line 317: a new variable $\alpha$ is introduced, but there exists no such variable in the G-BASE definition. What is this variable?
7. From the results, it looks like there is no straight-forward dependence on $K$. Large $K$ is not necessarily enabling better performance. Given this observation, is there a strategy to choose $K$?
8. The claim on the computational aspect of BASE and G-BASE is not substantiated by analysis (either theoretical or empirical). It is not immediately evident how BASE and G-BASE are computationally more efficient than other methods.
9. On the experimental details: what proportion of nodes are used for training each benchmark? Does it have an effect on the data population pool assumption? Adding this detail (to Appendix A) would be helpful.

Minor concerns (mainly to improve the paper quality):
1. Acronyms like BASE, G-BASE, LIRA, RMIA, and GCN should be clearly defined at first use. Given that this paper introduces BASE and G-BASE, are these acronyms?
2. Variable $\alpha$ in line 317 is not defined.
3. The AGGREGATE and UPDATE function in equation (1) should be further explained, especially for readers unfamiliar with GNNs. Adding references could be helpful.

**Ethical Concerns:**

["NO or VERY MINOR ethics concerns only"]

**Final Justification:**

UPDATE: The review addressed all my concerns. Hence, I am increasing my score. I will leave the decision on formatting on AC and beyond

**Limitations:**

The theoretically derived Bayes optimal rule is intractable and can only be approximately solved. This is not a major limitation, as is common in the literature.

**Paper Formatting Concerns:**

The authors left a comment ‘$i$ to $\tilde{M}$ - johan’ in line 228, which appears to reveal the author's identity and violates the double-blind reviewing policy.

**Quality:**

2

**Strengths And Weaknesses:**

Detailed comments are put under questions. Strengths and weakness are summarised below:
Strengths
1. Extending the Bayes optimal inference rule from [1]  to graphs is novel and important. The existing MIAs for graphs are only classifier-based methods.
2. The evaluation is extensive on the most common graph benchmarks and GNNs.


Weaknesses
1. Paper clarity can be improved, specifically in the technical aspects as mentioned in the questions.
2. The applicability of the baselines used for the analysis from the iid domain (LIRA and RIMA) to graphs is not clear.
3. CRITICAL: There is a potential formatting issue in 228, violating double-blind policy (see formatting issue below). My current score does not account for this violation, and I will let AC/SAC/PC decide on it

---

> ### Author Rebuttal · Authors · 2025-07-31
>
> We thank the reviewer for the thoughtful and detailed feedback, which helped us improve the paper. We believe we have addressed all of your comments thoroughly and satisfactorily. If our responses resolve your concerns, we would be grateful if you would consider updating your score accordingly.
>
> ## WEAKNESSES
>
> ### W1
> We appreciate the reviewer's feedback. We have addressed all questions raised in the review, and we believe our responses provide satisfactory clarification. We will update the technical sections in manuscript to reflect these clarifications and improve clarity.
>
> ### W2 and Q4.
> While LiRA and RMIA were originally developed for i.i.d. data, they can be adapted to the graph setting by treating the graph as a collection of node feature-label pairs and ignoring edge information during the attack. Specifically, the attack is applied to the “0-hop” neighborhood, where each target node is treated in isolation from its neighbors. Importantly, the shadow models are still trained on full graphs with connectivity, just like the target model.
>
> While one could attempt to use the full graph structure in the attack, i.e., obtain confidence values by querying the entire k-hop neighborhood, this typically leads to poorer performance, as the shadow graphs contain a mix of member and non-member nodes, making the model predictions on member and non-member nodes less distinguishable.
>
> We acknowledge that this adaptation was not clearly explained in the paper and will revise the manuscript to clarify this point.
>
> ### W3.
> We thank the reviewer for bringing this to our attention and sincerely apologize for the oversight on line 228, where the comment ``$i$ to $\tilde{\mathcal{M}}$ - johan'' containing a first name was inadvertently left in the submission. We fully recognize the importance of the double-blind review process and regret this unintentional mistake. However, we stress that a (common) first name alone is unlikely to identify the authors. We also note that, in practice, many submissions are available on arXiv, and author identities may be discoverable regardless. That said, we take this concern seriously and apologize again.
>
> ## QUESTIONS
> ### Q1.
> We thank the reviewer for catching this. There is indeed a typo in the argument of $S_\mathcal{L}$ in Equation~(8). The expression $S_\mathcal{L}(\phi_k, \mathcal{G}, m_v)$ should read $S_\mathcal{L}(f_{\phi_k}, v, \tilde{\mathcal{M}}_i, \mathcal{G})$. This is purely a typographical error and does not affect the derivation, implementation, or results in the paper. We will correct this in the camera-ready version, if accepted.
>
> ### Q2.
> There is indeed a notational oversight in Def. 2 of the G-BASE attack: the dependence on index $i$ is missing in $\tilde{\mathcal{M}}$. The correct expression is $\tilde{\mathcal{M}}_i$. This is a typographical issue only and does not affect the derivation or implementation, which correctly reflect the sampling procedure. We will fix the notation.
>
> ### Q3.
> We thank the reviewer for this comment. In our setting, the prior $p(\theta|\mathcal{G})$ (implicitly defined by the threat model) is the distribution of the model parameters before conditioning on $\tilde{\mathcal{M}}$, where $\tilde{\mathcal{M}}$ defines the subset used to train the model disregarding the target node. The prior distribution is then computed by integrating over subsets of $\mathcal{G}$, possibly taking advantage of any side-knowledge about the fraction of training nodes, the sampling procedure, or the training algorithm.
> In practice, this integration is estimated through Monte Carlo sampling, with samples obtained by training shadow models on subgraphs sampled from $\mathcal{G}$, which are independent of the target node and the $\tilde{\mathcal{M}}$ in the outer expectation. While this is described implicitly in Section 4.3, we acknowledge that this could have been made more explicit. We will revise the manuscript to clarify this point.
>
> We also note that assuming perfect knowledge of the training procedure is standard in the MIA literature, as it yields a meaningful upper bound on attack performance. Our mismatched adversary experiments (see Figure 9 in Section 6 and Table 8 in  Appendix H.2) specifically aim to test robustness when this assumption is relaxed.
>
> ### Q5.
> Our framework involves two distinct types of sampled graphs. The first corresponds to the subgraphs used to train the shadow models: these are used to approximate the inner expectation in both G-BASE (Definition 2) and BASE (Definition 3). Their number, denoted by $K$, is always equal to the number of shadow models, and this holds for all attacks. The second type refers to the sampled graphs used to compute the outer expectation in G-BASE (over membership configurations $\tilde{\mathcal{M}}$). The number of these samples, denoted by $M$, is an independent hyperparameter of the attack and does not need to match the number of shadow models. Both types of sampled graphs serve to approximate expectations via Monte Carlo estimation. Therefore, it is generally beneficial to use as many samples as possible, subject to the available computational budget.
>
> ### Q6.
> Variable $\alpha$ (referred to in line 317 of the paper) is a hyperparameter introduced in the offline version of BASE and G-BASE to account for the absence of shadow models trained with the target node. This compensates for the lack of target-specific information available in the offline setting. The same idea is used in RMIA, which also introduces a similar hyperparameter for the offline case.
>
> Due to space constraints, the discussion of the offline vs. online setting and the role of $\alpha$ was moved to App. F, which is referenced in the main text following line 299. For the reviewer's reference, we copy here the text following line 299:
>
> "**Online and offline setting.** We evaluate both online and offline attacks, as defined in Appendix F, and compare BASE and G-BASE with LiRA and RMIA in both settings."
>
> Appendix F clearly defines both settings and explains the introduction of $\alpha$ in the offline setting. We copy part of the text here for reference:
>
> "To compensate for the lack of in-models in the offline setting, we introduce a scaling factor $\alpha\in[0,1]$ on the Monte Carlo loss term over shadow models[…]”
>
> (see Eq. 22 in App. F)
>
> To improve clarity, we will explicitly add ``(see App. F)'' after mentioning $\alpha$ in the main text in the revised version of the manuscript.
>
> ### Q7.
> $K$ denotes the number of shadow models. As expected, increasing $K$ generally improves attack performance for all attacks, as it leads to a better approximation of the model distribution. However, this comes at the cost of training additional shadow models, which may be computationally expensive. In practice, $K$ is typically constrained by available computational resources. Importantly, the performance gains from increasing $K$ often exhibit diminishing returns beyond a certain point. BASE, G-BASE, and RMIA perform strongly even with a small number of shadow models; in some cases, we observe little to no improvement when increasing $K$ from 8 to 128. Hence, $K$ can be chosen to be small. LiRA, on the other hand, tends to benefit more from larger $K$ within this range, although prior work suggests that its performance also eventually saturates [1, Fig. 17]. A key advantage of our attacks, as well as RMIA, is that they typically achieve strong performance even with a low number of shadow models.
>
> [1] Carlini, et al. "Membership inference attacks from first principles." SP, 2022.
>
> ### Q8.
> Due to space constraints in the rebuttal, we refer to our response to Q2 from Reviewer U2Mu, which raised a similar question. As both questions are closely related, we chose to provide a unified answer there.
>
> ### Q9.
> Except in the mismatched adversary setting, all experiments use 50% of the nodes to train both target and shadow models; this will be clarified in Appendix A. In the mismatched (Sec. 6, following line 341), the challenger trains on 35% of the nodes, while the adversary assumes 50%, creating a deliberate mismatch.
>
> The data population pool assumption is flexible and allows for any proportion of nodes to be used when training target and shadow models. While a more realistic and challenging setting may involve the challenger training on a smaller subset of a larger  graph, using 50% of the nodes provides a practical and meaningful baseline when auditing GNNs on a specific dataset.
>
> In a privacy auditing scenario, where a model provider wants to measure the privacy of a model that will be trained on a dataset, the target model to audit has to be trained on a fraction of the dataset to leave out non-member samples. Using a too small fraction can lead to models that differ significantly from those trained on the full dataset, hence MIAs may fail to properly measure the privacy risk of the full model. Conversely, using a very large training fraction, and assuming a “matched'' adversary, can result in substantial overlap between the shadow and target training sets, leading to overly potent attack performance. Using 50% strikes a reasonable balance: it avoids unrealistic overlap while ensuring the model is representative of typical training conditions.
>
> ## MINOR CONCERNS
> ### MC 1.
> We defined G-BASE (Graph Bayes-Approximate membership Status Estimation) in Line 299, after the formal definition of the attack. However, we agree it would be clearer to introduce it at first mention and will move the definition accordingly. Note that the acronym omits the “M” in “membership” to align phonetically with BASE, which is intended to evoke “Bayes.”
>
> The name BASE follows the same naming convention, but we did not define it explicitly. We will revise the manuscript and define all acronyms.
> ### MC 2.
> See our response to your Q6.
> ### MC 3.
> We will add a brief explanation of these functions and include an additional reference to improve clarity for readers less familiar with GNNs.

---

> > ### Comment · Area_Chair_Bzjh · 2025-08-05
> >
> > Dear reviewer CHe9,
> >
> > As the author–reviewer discussion period comes to a close, please take some time to carefully read and respond to the authors’ rebuttal. It is important to engage in the discussion with the authors before acknowledging that you have read the rebuttal.
> >
> > Regards, \
> > AC

---

### Official Review · Reviewer_g4At · 2025-07-04

**Clarity:** 3
**Significance:** 3
**Originality:** 2
**Rating:** 4
**Confidence:** 2

**Summary:**

This paper proposes a novel and low-cost node-level membership inference attack for graph neural networks. The authors further extend their method to the i.i.d. data setting and establish a theoretical connection between their approach and existing methods. Experiments across six datasets demonstrate that the proposed method achieves comparable or superior performance to existing attacks under the same parameter configurations, while maintaining lower computational cost and higher robustness.

**Questions:**

This paper presents a promising and practical approach for membership inference, particularly on GNNs. The extensions to the i.i.d. setting and the theoretical justification of the proposed approximations are valuable contributions.
However, I have a few concerns and suggestions for improvement:
1.	In Section 6, while the experiments are comprehensive, the analysis lacks depth. There should be more deeper discussion, for example, why the proposed methods perform better on some datasets but not on others, and why your methods have better robustness against mismatched adversarial assumptions.
2.	In Section 4.3, several sampling strategies are proposed to approximate the Bayes-optimal rule. It would be helpful to clarify how each strategy influences performance, and which specific strategies are used in your experiments.
3.	The organization of the paper could be improved. Sections 2 to 5 could be made more concise to allow more space for in-depth experimental discussion and analysis in Section 6.

**Ethical Concerns:**

["NO or VERY MINOR ethics concerns only"]

**Final Justification:**

Based on the additional experiments provided by the authors, I still choose to maintain my positive score.

**Limitations:**

yes

**Paper Formatting Concerns:**

.

**Quality:**

2

**Strengths And Weaknesses:**

Strengths:
⦁	Better or equal performance with existing methods but with lower cost.
Weaknesses:
⦁	The core idea of this paper is Bayes-optimal framework, which has been previously explored on MIAs, limiting the novelty.
⦁	Writing quality needs improvement for clarity and precision.

---

> ### Author Rebuttal · Authors · 2025-07-30
>
> We thank the reviewer for the thoughtful and detailed feedback, which has helped us improve the paper. We believe we have addressed all of your comments thoroughly and satisfactorily. If our responses resolve your concerns, we would be grateful if you would consider updating your score accordingly.
>
> ## WEAKNESSES:
>
> *W1. "The core idea of this paper is Bayes-optimal framework, which has been previously explored on MIAs, limiting the novelty."*
>
> ### Response:
> While our work builds on the Bayes-optimal framework introduced in [1], we make substantial and novel contributions by extending it to graph neural networks (GNNs). Specifically, we derive the Bayes-optimal decision rule for node-level MIAs against GNNs, revealing how graph structure fundamentally impacts optimal attack strategies, an aspect not addressed in previous work.
>
> We also derive practical approximations to the Bayes-optimal decision rule for the i.i.d. case, leading to attacks that not only outperform those in [1], but also match or surpass the state-of-the-art attacks for i.i.d. data, namely,  LiRA and RMIA.
>
> Finally, we formally prove the equivalence between our attack BASE and RMIA, providing a rigorous foundation for a widely-used state-of-the-art method.
>
> Together, these contributions represent a significant advancement of both the theory and practice of MIAs.
>
> [1] Sablayrolles et al. "White-box vs black-box: Bayes optimal strategies for membership inference." ICML, 2019.
>
> *W2. "Writing quality needs improvement for clarity and precision."*
>
> ### Response:
> We appreciate the feedback and acknowledge that there are a few typos and minor phrasing issues, which we will correct in the final version. We have aimed to present our ideas as clearly and precisely as possible, and some reviewers found the writing to be strong. We will continue to refine the presentation to ensure maximum clarity.
>
> ## QUESTIONS:
>
> *Q1. "In Section 6, while the experiments are comprehensive, the analysis
> lacks depth. There should be more deeper discussion, for example, why the
> proposed methods perform better on some datasets but not on others, and
> why your methods have better robustness against mismatched adversarial
> assumptions."*
>
> ### Response:
> We agree that a more detailed interpretation of the experimental results would strengthen the paper. If accepted, we will revise the discussion accordingly in the camera-ready version, using the additional page. Below, we highlight some insights:
>
> G-BASE performs worse on smaller and sparser datasets like Cora and Citeseer. In our setup, target and shadow graphs are constructed by sampling 50\% of the nodes, which significantly reduces graph connectivity and can lead to isolated nodes. Since G-BASE relies on neighborhood structure to infer membership, its effectiveness degrades when this structure is weak or fragmented. In contrast, on larger and denser datasets, where the sampled subgraphs retain more structural information, G-BASE can better exploit the graph topology and achieves stronger performance. We sample 50% of the datasets 10 times and report the average degree of a node (the number of edges connected to the node) and the fraction of isolated nodes:
>
> | | Average degree | Fraction of isolated nodes (%) |
> |----------|:----------:|:----------:|
> | Cora  | 1.9523  | 18.14 |
> | Citeseer | 1.3293 | 30.81  |
> | Pubmed | 2.2375 | 28.90 |
> | Flickr | 5.0215 | 1.63 |
> | Amazon-Photo | 15.4271 | 4.02 |
>
> This reveal that the citation network become fairly sparse after sampling 50% of the nodes uniformly. However, G-BASE achieves comparatively good performance on Pubmed despite its many isolated nodes and high fraction of isolated nodes, indicating that this is not the full story. To gain further insights, we plan to investigate the properties and local neighborhood for the nodes that the attacks identify as members with the highest confidence.
> (The Github dataset also presented in the paper is currently unavailable to us due to a bug in PyG. However, since the Github dataset has 37700 nodes and 578006 edges, it is likely to retain high connectivity even after sampling 50% of the nodes uniformly).
> We will revise the manuscript to include a more thorough discussion around these points. However, we stress that explaining the behaviour of MIAs against GNNs is further complicated by the complex interactions between nodes that are introduced by the message-passing mechanism. Therefore, it is non-trivial to provide further insights into the behaviour of the attacks.
>
> As for the robustness under mismatched adversarial assumptions, we do not yet have a definitive explanation for why some attacks, including ours, exhibit greater robustness. However, we note that this remains an open question in the literature; similar observations were made but not fully explained in both LiRA [2] and RMIA [3] papers for i.i.d. data.
>
> [2] Carlini, Nicholas, et al. "Membership inference attacks from first principles." SP, 2022.
>
> [3] Zarifzadeh et al. "Low-cost high-power membership inference attacks", ICML 2024.
>
> *Q2. "In Section 4.3, several sampling strategies are proposed
> to approximate the Bayes-optimal rule. It would be helpful to clarify how
> each strategy influences performance, and which specific strategies are used
> in your experiments."*
>
> ### Response:
> The influence of each sampling strategy on G-BASE performance is evaluated and compared in Appendix H.3, titled "Comparison of Sampling Strategies for G-BASE."
>
> In the main paper, the specific sampling strategy used in each experiment is indicated by a label: (MI) denotes model-independent sampling, and (MIA) refers to 0-hop MIA sampling. For the reviewer's convenience, we provide an excerpt from Section 6:
>
> "We evaluate G-BASE for both model-independent sampling (MI) and $0$-hop MIA sampling (MIA) (see Section 4.3), using BASE as the $0$-hop MIA. In offline mode, only the best-performing sampling strategy is reported.
> Results with MCMC sampling are presented in Appendix H.3."
>
> and another excerpt from Appendix H.3:
>
> "In Section 4.3 and Appendix B.2, we proposed three different sampling strategies to sample from $P(\tilde{\mathcal{M}}|\theta,\mathcal{G})$, for the purpose of evaluating the expected value in the Bayes-optimal membership inference rule.
> Here, we evaluate and compare the impact of the sampling strategy on the attack performance of G-BASE.
> Table 9 shows the attack performance of G-BASE over different datasets and model architectures, using each of our three sampling strategies: model-independent sampling (MI), 0-hop MIA sampling (MIA), and the Metropolis-Hastings sampling (MCMC)."
>
>
>
> *Q3. "The organization of the paper could be improved.
> Sections 2 to 5 could be made more concise to allow more space for in-depth
> experimental discussion and analysis in Section 6."*
>
> ### Response:
> We appreciate the suggestion. We agree that expanding the experimental analysis would strengthen the paper. If accepted, we will use the additional page in the camera-ready version to provide a deeper discussion of the results in Section 6.

---

> ### Comment · Reviewer_g4At · 2025-08-07
>
> Thank you for the authors’ response, which has addressed my concerns. I look forward to seeing the full version of the paper.

---

### Official Review · Reviewer_L9uX · 2025-07-04

**Clarity:** 3
**Significance:** 2
**Originality:** 3
**Rating:** 4
**Confidence:** 2

**Summary:**

This paper introduces BASE and G-BASE, two efficient approximations of Bayes-optimal MIA attacks for standard and graph data. Their method BASE matches the baseline performance with far fewer queries and the graph version also outperforms previous baselines. The work is theoretically sound with experiments, but only on small datasets, casting a doubt on real world scalability.

**Questions:**

1. Can this be verified on reasonably large datasets like ImageNet? Example tasks can be generation models trained on the dataset.
2. Another prohibiting factor for real world deployment is the cost of querying these large models for obtaining log-probs. Would love to hear authors' take on that.

**Ethical Concerns:**

["NO or VERY MINOR ethics concerns only"]

**Final Justification:**

Though not fully convinced by the rebuttals, I understand their point about the state of the field and existing standard datasets.
In that purview, I retain my borderline accept for this work.

**Limitations:**

yes

**Quality:**

2

**Strengths And Weaknesses:**

Strengths:
- The paper is well written with solid grounding in theory and backed by rigorous empirical methods.
- They introduce a novel unified framework for Graph-data and other forms of data.

Weaknesses:
- Though the authors have shown excellent results, the scale of the datasets used question the real world use cases. Can Bayes-optimal MIA scale to real world models like SD-3.5 or FLUX image gen models which are trained across billions of image-text pairs ? It is hard to draw conclusions about the practicality when the empirical data is orders of magnitude smaller.
- In the same vein, the method also requires training multiple shadow models which is also not viable as training good large models costs millions of dollars.

---

> ### Author Rebuttal · Authors · 2025-07-30
>
> We thank the reviewer for the thoughtful and detailed feedback, which has helped us improve the paper. We believe we have addressed all of your comments thoroughly and satisfactorily. If our responses resolve your concerns, we would be grateful if you would consider updating your score accordingly.
>
> ## WEAKNESSES:
>
> Note: We provide a joint response to Reviewer’s Weaknesses 1 and 2, as they are closely related.
>
> *W1 and W2. "Though the authors have shown excellent results, the scale of the datasets used question the real world use cases. Can Bayes-optimal MIA scale to real world models like SD-3.5 or FLUX image gen models which are trained across billions of image-text pairs? It is hard to draw conclusions about the practicality when the empirical data is orders of magnitude smaller. In the same vein, the method also requires training multiple shadow models which is also not viable as training good large models costs millions of dollars."*
>
> ### Response:
> Our work focuses  on models trained to perform classification tasks, as is standard in the MIA literature (e.g., LiRA and RMIA). While our Bayes-optimal MIA framework can be extended to regression models and even autoregressive LLMs, it is not directly applicable to generative diffusion models such as SD-3.5 and FLUX. Deriving a Bayes-optimal MIA against such models is an interesting but separate research direction.
>
> The datasets we consider are standard in the GNN literature, and more diverse than what has previously been explored in node-level MIA work [2-4]. Furthermore, our datasets include real-world examples such as Amazon-Photo [5] which is a segment of the Amazon-co purchase graph [6] containing objects from the Amazon web store. Thus, the experiments we consider also reflects realistic use cases.
>
> Regarding scalability, state-of-the-art MIAs, including LiRA, RMIA, and ours, require training shadow models, which becomes prohibitively expensive for models with billions of parameters and training examples. Our method is no exception but neither are the baselines LiRA and RMIA. There is a recent work that scales up LiRA to moderately-large language models [1], but this requires computational resources well beyond what we have access to. Ultimately, attacking models that are extremely costly to train is itself extremely costly.
>
> [1] Hayes et al. "Strong Membership Inference Attacks on Massive Datasets and (Moderately) Large Language Models." arXiv, 2025.
>
> [2] Olatunji et al., "Membership Inference Attack on Graph Neural Networks" TPS-ISA, 2021.
>
> [3] Duddu et al., "Quantifying Privacy Leakage in Graph Embedding," MobiQuitous, 2020.
>
> [4] He et al., "Node-Level Membership Inference Attacks Against Graph Neural Networks," arXiv, 2021.
>
> [5] Shchur et al., "Pitfalls of Graph Neural Network Evaluation," NeurIPS, 2018.
>
> [6] McAuley et al., "Image-based Recommendations on Styles and Substitutes," SIGIR, 2015
>
> ## QUESTIONS:
>
> *Q1. "Can this be verified on reasonably large datasets like ImageNet? Example tasks can be generation models trained on the dataset."*
>
> ### Response:
> BASE can indeed be applied to ImageNet-trained classifiers, and a comparison with other i.i.d. MIAs on ImageNet is possible. However, crucially, we prove in the paper that BASE is equivalent to RMIA (Theorem 2), which was evaluated on ImageNet in its original paper  [7]. Thus, due to our theoretical result, BASE is guaranteed to match RMIA's strong performance on ImageNet, at a reduced computational cost due to needing fewer model queries. Given this theoretical equivalence, we believe additional experiments on i.i.d. datasets would offer limited new insight. Instead, we focus more on graph data, where our contributions and experiments are extensive.
>
> We also reiterate that our Bayes-optimal MIA is designed for classification tasks (as it is standard in the MIA literature) and does not directly apply to generative models.
>
> [7] Zarifzadeh et al., "Low-cost high-power membership inference attacks.", ICML, 2024.
>
> *Q2. "Another prohibiting factor for real world deployment is the cost of querying these large models for obtaining log-probs. Would love to hear authors' take on that."*
>
> ### Response:
> We agree that minimizing the number of queries is critical in real-world scenarios, where API access may be limited or costly. Our attack BASE requires only a single query to the target model, yet matches the performance of RMIA, which can require thousands of queries. Our G-BASE attack requires just two queries to the target model per sampled graph. As our results indicate, 8 sampled graphs yields already strong performance, and so 16 queries is enough (per target sample).
>
> Moreover, querying GNNs (our focus) is significantly cheaper than querying large generative image/video models.
>
> Finally, in the MIA literature, it is common to assume that the black box access to the target model produces class probabilities (e.g., LiRA and RMIA papers), which is also the setting we study. There are "label-only" attacks that weaken this assumption and assume only access to the predicted label, but this falls outside the scope of this work. In practice, APIs to models can give either soft or hard predictions, depending on the use-case, and so both scenarios are interesting to study.

---

> > ### Author Response · Authors · 2025-08-06
> >
> > Dear Reviewer,
> >
> > We believe we have thoroughly and satisfactorily addressed all of your comments in our response. We would greatly appreciate it if you could take a moment to review our replies and let us know if any concerns remain. We're happy to clarify any points further.
> >
> > Thank you,
> >
> > The authors

---

### Note · Authors · 2025-08-11

**Summary**: The original reviews were overall positive: L9uX (score 4), g4At (score 4), and U2Mu (score 5) all expressed appreciation for our contribution. CHe9 (3, weak reject) was the only critical reviewer.

**Discussion outcome**:

For L9uX, their remarks focused on scalability and the cost of querying a model. We believe we addressed these points thoroughly and convincingly, although they did not engage in discussions.

For g4At, their main remarks concerned the need for a deeper discussion of the results and strategies. We addressed their comments in detail, and they explicitly expressed satisfaction with our rebuttal.

For U2Mu, the reviewer asked about the justification of our threat model, additional explanations, and suggested a further discussion around the computational cost and societal impact. We provided detailed responses that we believe addressed these points satisfactorily.

For CHe9, we addressed each of their remarks in detail and believe our responses warrant a careful reconsideration of their score. However, CHe9 did not provide any follow-up.

**More precisely for CHe9**:

Most of Rev CHe9’s points were minor. Q1 and Q2 concerned typographical issues. Q6 regarded a point they believed were not discussed in the paper but in fact were covered (we pointed to the explanations in the paper and also provided further clarifications that we will include in the revised manuscript). Q3, Q4, Q5 and Q7 involved providing further explanations about technical details of our approach. We clarified all of these and will update the paper accordingly.

The main point raised by CHe9 (Q8) was to provide stronger support for our computational efficiency claim (this point was also raised by U2Mu, who nevertheless gave the paper a good score).

In the paper, we supported this claim by discussing that BASE requires significantly fewer model queries than the prior SOTA RMIA (which we prove have equivalent attack performance). Fewer queries make the attack more efficient, as querying large models can be expensive. We also supported via experiments that BASE requires fewer shadow models than LiRA to achieve similar strong performance.

Following the reviewers’ comments, we further strengthened our claims by measuring wall-clock inference times (as proposed by U2Mu) for BASE compared to LiRA and RMIA. The results show that BASE is significantly faster in practice. We will include this discussion in the manuscript, and we believe that it strengthens the paper.

---

### Decision · Program_Chairs · 2025-09-17

**Decision:**

Accept (poster)

**Comment:**

This paper studies the problem of MIA in the node-level graph data setting. Following the prior Bayesian MIA framework, the authors develop more efficient MIA methods compared with existing methods.

The reviewers agree the problem considered in this paper is important, and proposed methods indeed work well. While concerns remain about task coverage, data size and scalability of the methods, the reviewers believe the paper advances MIA on graph data.

Therefore, I recommend the weak accept of the paper.